# Double boron–oxygen-fused polycyclic aromatic hydrocarbons: skeletal editing and applications as organic optoelectronic materials

Guijie Li [1] ✉, Kewei Xu[1], Jianbing Zheng[1], Xiaoli Fang[1], Yun-Fang Yang [1], Weiwei Lou[1], Qingshan Chu[1], Jianxin Dai[1], Qidong Chen[1], Yuning Yang[1] & Yuan-Bin She[1] ✉

An efficient one-pot strategy for the facile synthesis of double boron–oxygen-fused polycyclic aromatic hydrocarbons (dBO-PAHs) with high regioselectivity and efficient skeletal editing is developed. The boron–oxygen-fused rings exhibit low aromaticity, endowing the polycyclic aromatic hydrocarbons with high chemical and thermal stabilities. The incorporation of the boron–oxygen units enables the polycyclic aromatic hydrocarbons to show single-component, low-temperature ultralong afterglow of up to 20 s. Moreover, the boron–oxygen-fused polycyclic aromatic hydrocarbons can also serve as ideal $n$-type host materials for high-brightness and high-efficiency deep-blue OLEDs; compared to single host, devices using boron–oxygen-fused polycyclic aromatic hydrocarbons-based co-hosts exhibit dramatically brightness and efficiency enhancements with significantly reduced efficiency roll-offs; device 9 demonstrates a high color-purity (Commission International de l'Eclairage $CIE_y = 0.104$), and also achieves a record-high external quantum efficiency (28.0%) among Pt(II)-based deep-blue OLEDs with Commission International de l'Eclairage $CIE_y < 0.20$; device 10 achieves a maximum brightness of 27219 cd/m$^2$ with a peak external quantum efficiency of 27.8%, which representes the record-high maximum brightness among Pt(II)-based deep-blue OLEDs. This work demonstrates the great potential of the double boron–oxygen-fused polycyclic aromatic hydrocarbons as ultralong afterglow and n-type host materials in optoelectronic applications.

Polycyclic aromatic hydrocarbons (PAHs), especially heteroatom-embedded PAHs, have attracted considerable attention[1] because of their unique catalytic activity and photophysical/photoelectronic properties, enabling extensive and promising applications in catalysis[2] and organic electronics, such as organic photovoltaics (OPVs)[3], organic field-effect transistors (OFETs)[4,5], organic phototransistors (OPTs)[6], and organic light-emitting diodes (OLEDs)[7–11]. However, because of strong vibronic coupling and aromaticity, PAHs typically show broad structural emission spectra and low T$_1$ energy levels ($E_{T1}$) (Supplementary Fig. 1)[9]. Among the various main-group elements used

[1]College of Chemical Engineering, State Key Laboratory Breeding Base of Green-Chemical Synthesis Technology, Zhejiang University of Technology, 310014 Hangzhou, Zhejiang, P. R. China. ✉e-mail: guijieli@zjut.edu.cn; sheyb@zjut.edu.cn

as dopants, boron is particularly attractive, owing to its sp[2] hybridization and the presence of the vacant $p_z$ orbital; these features enable the efficient regulation of the molecular skeletons and π-conjugation of the boron-embedded PAHs (B-PAHs), as well as the further tuning of their excited-state properties and charge-transfer characteristics, in order to develop a variety of novel functional materials[1,7–13]. In particular, boron/nitrogen-embedded PAHs (BN-PAHs) with rational molecular design demonstrated extremely narrowband Gaussian-type emission spectra because of the minimized vibronic coupling by multiple resonance (MR) effect[7–9]. However, compared with the widely studied B-PAHs and BN-PAHs[1,6–17], scarce data are available on PAHs containing B–O unit (BO-PAHs), and in particular on their optoelectronic properties, due to their inefficient synthetic methods.

Since Dewar and Dietz's pioneering work on the synthesis of 9,10-boroxarophenanthrene in 1960[18], several groups made good progress on investigating the chemistry of BO heteroarenes in recent years (Fig. 1a)[19–23]. However, some issues still exist, such as harsh reaction conditions, separation of air-sensitive intermediates, or low isolated yields. Thus, the development of facile synthetic methods, which can facilitate the synthesis of chemically diverse BO-PAHs and the systematic investigation of their structure-property relationship, is still highly desired. These are fundamental and critical targets for both chemists and material scientists, whose achievement would also be beneficial to promote the applications of the BO-PAHs as functional materials.

Herein, we report an efficient method for the facile fabrication of doubly BO-fused PAHs (dBO-PAHs) with versatile molecular skeletons via a one-pot strategy of tandem demethylation-electrophilic borylation-nucleophilic substitution reactions (Fig. 1b). The molecular cores of the dBO-PAHs could be efficiently edited with high regioselectivity by regulating the positions of the aryl (Ar[1] and Ar[2]) and methoxyl (OMe) groups in the phenyl rings of the substrates; the aryl and OMe moieties acted as guiding and directing groups, respectively. Using this method, doubly BO-fused central symmetric pyrenes (Type I) and benzo[m]tetraphenes (Type III and Type VI), $C_2$-symmetric benzo[k] tetraphenes (Type IV and Type V), and asymmetric benzo[c]chrysene (Type II) were successfully synthesized (Fig. 1b). Their molecular geometries, photophysical and thermal properties were systematically investigated through crystallographic studies, optical spectroscopy, and thermal gravimetric analysis (TGA). Theoretical and experimental studies revealed that the incorporation of the B–O units into the PAHs greatly affected their aromaticity and electronic properties. The synergistic effects of the p-π conjugation, the delocalization of the lone electron pairs of the oxygen atoms as well as the bulky Mes moieties (Mes: 2,4,6-trimethylphenyl) enabled the dBO-PAHs to possess good stabilities. The dBO-PAHs showed strong fluorescence emission with quantum efficiencies ($\Phi_{PL}$) of up to 95% at room temperature (RT) in dichloromethane; moreover, most dBO-PAHs exhibit single-component, dual-emission with both nanosecond fluorescence and second-level phosphorescence at 77 K in 2-methyltetrahydrofuran (2-MeTHF), and the persistent phosphorescence enabled the dBO-PAHs to exhibit an ultralong low-temperature afterglow of up to 20 s after UV excitation. Furthermore, some dBO-PAHs could also serve as ideal n-type host materials for high-brightness and high-efficiency deep-blue OLEDs; compared to single host, **BO1b**-based co-host systems enabled deep-blue OLEDs to exhibit dramatic brightness and efficiency enhancements with significantly reduced efficiency roll-offs; PtON1 and PtON-TBBI-based OLEDs realized a record-high maximum brightness ($L_{max}$, 27219 cd/m²) and a record-high external quantum efficiency (EQE, 28.0%) among Pt(II)-based deep-blue OLEDs with Commission International de l'Eclairage (CIE) y < 0.20, respectively.

## Results
### Synthetic method development and structure characterization
To develop a facile synthetic method for dBO-PAHs, we envision a one-pot strategy to avoid the separation of the air-sensitive intermediates

(**Int 1** and **Int 3**) via tandem demethylation-electrophilic borylation-nucleophilic substitution reactions using dimethoxybiaryls or dimethoxytriaryls as starting materials (Fig. 2). However, the simultaneously form two B–O bonds and four B–C bonds remains challenging because of the complex reaction system involved; in particular, in the synthesis of dBO-PAHs with two boron atoms in the same phenyl ring, e.g., Type I, Type V, and Type VI compounds, it is difficult to attain the second electrophilic borylation on an electron-deficient phenyl ring (Fig. 2).

After optimizing the reaction conditions (see Supplementary Fig. 2 for details), it was found that pure BBr₃ could accelerate the demethylation; additionally, nonpolar solvents improved the solubility of the intermediates and enhanced the reaction rate. Finally, a 94% isolated yield of **BO1a** in gram scale could be achieved under mild conditions using hexane/toluene as a solvent in the presence of 2.5 equiv of pure BBr₃, 4–8 mol% AlCl₃ and 5.0 equiv of MesMgBr. Notably, the bulky Mes groups were incorporated to enhance the chemical stability by protecting the boron atoms from nucleophilic reagents[24].

As illustrated in Fig. 3, the one-pot protocol exhibited a broad substrate scope. For the synthesis of central symmetric double BO-fused pyrenes (Type I), electron-rich alkyl and aryl substitutions on either of the phenyl rings were tolerated in moderate to excellent yields (68–94%), however, 2.0 equiv AlCl₃ was needed for **BO1e** to promote the complete conversion of intermediate. Interestingly, upon adjusting the position of one phenyl group (Ar[2]) from 1,3-dimethyl-2,5-diphenylbenzene (**A1f**) to 1,3-dimethyl-2,4-diphenylbenzene (**A2**), the asymmetric dBO-benzo[c]chrysene **BO2** (Type II), bearing two boron atoms on different phenyl rings, could be exclusively separated in 80% yield, and the corresponding Type I compound was not observed; this revealed that the first borylation was dominated by the steric effect, and the second electrophilic borylation had a high regioselectivity and preferentially occured on an electron-rich phenyl (Fig. 1b). By further adjusting the Ar[1] position of **A2**, another series of dimethoxybiaryls **A3** were designed, which generated dBO-benzo[m]tetraphenes (Type III) with central symmetric skeletons (Fig. 1b). Notably, this reaction system also tolerated β-naphthyl and 2-fluorenyl groups, forming the extended π-conjugation molecules **BO3d** and **BO3e** with polygonal-shaped skeletons in 79% and 41% yields, respectively (Fig. 3). On the other hand, by adjusting both OMe and Ar[2] positions of **A2**, the obtained **A4** smoothly generated $C_2$-symmetric Type IV **BO4a**–**BO4e** in 43–85% yields, in which methyl, sterically hindered α-naphthyl, β-naphthyl, and 2-fluorenyl groups were all tolerated (Figs. 1b, 3). Unexpectedly, the electrophilic borylations exclusively occurred on the relatively low-reactivity β-positions of the naphthyl groups in **BO4d** (Fig. 3), revealing that steric effects prevailed over electronic effects. Importantly, the positions of the boron atoms in the molecular skeletons could be effectively edited by simply adjusting the OMe position of **A4** and **A3** to produce the corresponding $C_2$-symmetric **BO5** (Type V) and central symmetric **BO6** (Type VI) with high regioselectivities in 83% and 81% yields, respectively (Figs. 1b, 3); the regioselectivities were attributed to the steric effects of the Mes groups, which impeded the last nucleophilic substitution reaction to form isomers. Notably, all attempts to introduce tert-butyl into the Type I and Type III BO-PAHs failed, and only complete de-tert-butylation products of retro-Friedel–Crafts reactions were isolated[25] owing to the strong Lewis acidity of AlCl₃ (Supplementary Fig. 3). Ortho-analogs were also tried to synthesized by using the precursors of 1,4-dibromo-2,3-dimethoxy-5,6-dimethylbenzene (**A7a**) and 2,3-dimethoxy-1,4-diphenylnaphthalene (**A7b**), but they gave messy results (see supporting information for details). The dBO-PAHs were highly stable and could be easily purified through column chromatography on silica gel. Importantly, they were also insensitive to air and moisture, and no degradation was observed after keeping them in the solid state

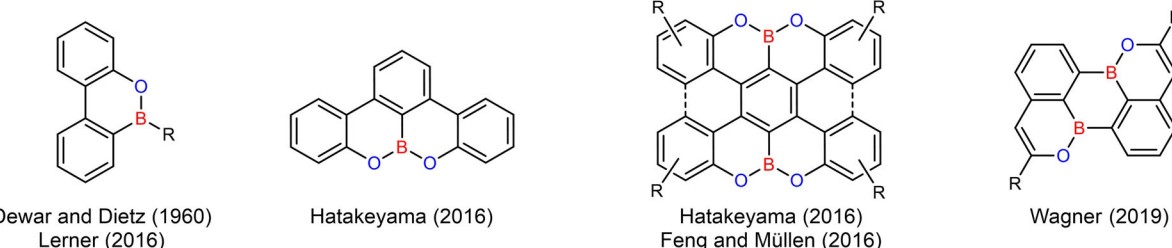

**a** ■ Classical BO-PAHs: BO-phenanthrene, OBO-benzo[*fg*]tetracene, OBO-bistetracene and BO-perylene

Dewar and Dietz (1960)
Lerner (2016)

Hatakeyama (2016)

Hatakeyama (2016)
Feng and Müllen (2016)

Wagner (2019)

**b** ■ Facile synthesis of double BO-fused PAHs with easy skeletal editing through one-pot strategy (this work)

**one-pot:** (i) **B**Br₃ (2.5 equiv), hexane/toluene, r.t., 12-40 h; (ii) AlCl₃ (4-8 mol%), r.t.-75 °C, 8-10 h; (iii) MesMgBr (5.0 equiv), r.t., 1-10 h.

✓ easy access to starting materials ✓ facile skeletal editing ✓ single-component dual-emissive materials
✓ one-pot tandem synthesis ✓ insensitive to air and moisture ✓ ultralong low-temperature phosphors ($\tau_p$ up to 5.0 s)
✓ excellent regioselectivity ✓ high thermal stability ✓ efficient room-temperature emitters ($\Phi_{PL}$ up to 95%)
 ✓ good *n*-type host material for a record-high $L_{max}$ and a record-high EQE$_{max}$ among Pt(II)-based deep-blue OLEDs

**Fig. 1 | Schematic illustration of the current work. a** Previous reported BO-PAHs. **b** Design concept of versatile skeletal editing and facile synthesis of dBO-PAHs through one-pot strategy developed in this work. The versatile skeletal editing can be easily realized by Ar¹, Ar² or OMe position regulation; the arrows indicate the position of electrophilic borylation directed by the OMe group. Ph, phenyl group, Mes, 2,4,6-trimethylphenyl group, Ar¹, Ar², aryl group, R, R¹, R², substituent group.

under ambient conditions for more than 1 year. All the newly developed dBO-PAHs were characterized by ¹H, ¹³C, and ¹¹B nuclear magnetic resonance (NMR) spectroscopy and high-resolution mass spectrometry (HRMS), and the results were consistent with the proposed molecular structures. Importantly, large-scale synthesis of **BO1b**, **BO1c**, **BO2,** and **BO3a** were also carried out for their further device fabrication as host materials; they all could be smoothly obtained using 4 mol % AlCl₃ with isolated yields of 80% (9.56 g), 87% (16.30 g), 80% (6.71 g) and 82% (10.98 g), respectively. The developed method can provide versatile dBO-PAHs, enabling

**Fig. 2 | Reaction pathway.** Plausible reaction pathway for the one-pot synthesis of dBO-PAHs.

the investigation of their potential applications in organic electronics.

## Crystallographic analyses

To further confirm the molecular structures of the dBO-PAHs and determine their molecular geometries, single crystals of **BO1a, BO2, BO3a, BO3d, BO4b, BO4d, BO4e, BO5,** and **BO6** were carefully grown and analyzed by X-ray diffraction. The crystal structures, selected bond lengths, bond angles, dihedral angles, and solid-state packing structures are illustrated in Fig. 3, Fig. 4, and Supplementary Figs. 4–14, and the crystal data are listed in Supplementary Tables 1–4 and Supplementary Data 1–10. The crystal diffraction analyses clearly support the molecular structures obtained from NMR and HRMS data (Fig. 3). All the dBO-PAHs had almost planar BO-fused cores, supported by the fact that the sums of the one C–B–C and two C–B–O bond angles around the boron atoms were almost 360.0° for all the dBO-PAHs, revealing the sp$^2$ hybridization of the central boron atoms. However, **BO3a** exhibited a slight bent core with a dihedral angle of 10.33° between the two terminal phenyl rings (Supplementary Fig. 8). Moreover, the B–C bond lengths of the BO-cores in **BO1a, BO2, BO4b, BO4d, BO4e, BO5,** and **BO6** ranged between 1.522 and 1.549 Å; these values were significantly shorter than those of their B–C$_{Mes}$ bond lengths (1.564–1.578 Å) and the reported B–C bond lengths of Mes$_3$B (1.573–1.580 Å)[26]. The short B–O bond lengths are from 1.368 Å to 1.384 Å, which are in line with previous reports[19,21,22]. These results reveal that the empty boron p$_z$ orbitals were stabilized not only by the strong p-π conjugation with adjacent phenyl rings, but also by the delocalization effect of the lone electron pairs from the directly linked oxygen atoms, endowing the dBO-PAHs with excellent stabilities. The p-π conjugation and the delocalization effect were also supported by the $^{11}$B NMR spectra of the dBO-PAHs, which showed significant upfield shifts ($\delta$ = 44.68–49.59 ppm) compared to that of the Mes$_3$B ($\delta$ = 79.0 ppm)[27]. These results reveal the importance of the enforced planarity of the BO cores for the fabrication of stable dBO-PAHs. Furthermore, as expected, the BO cores were almost spatially perpendicular to the bulky Mes groups, enabling the boron atoms to be shielded by the methyl groups in the *ortho* position of the Mes moieties (Supplementary Fig. 5), which was also beneficial for the stability enhancement.

All the dBO-PAHs had monoclinic crystal systems, except **BO2** and **BO4b**, which crystallized in orthorhombic systems (Supplementary Tables 1–4). Continuous π-stacks with varying degrees of slipped stacking between the BO cores were observed in all dBO-PAHs except **BO6** (Fig. 4, and Supplementary Figs. 6–13). **BO3a** and **BO4b** showed small interplanar distances of 3.446 and 3.685 Å, respectively, indicative of strong intermolecular π⋯π interactions (Fig. 4, and Supplementary Figs. 8, 10). In contrast, **BO2, BO3d,** and **BO4e** exhibited large interplanar distances of 5.967, 6.404, and 5.536 Å, respectively (Supplementary Figs. 7, 9, 12). Notably, **BO1a** showed two types of π-stacks in almost perpendicular directions, with interplanar distances of 3.619 and 6.092 Å (Fig. 4, and Supplementary Fig. 6). On the other hand, intermolecular C–H⋯π interactions between the Mes groups were also observed for **BO3a, BO4b,** and **BO4d** with distances of 3.183, 2.859, and 2.883 Å, respectively (Supplementary Figs. 8, 10, 11). Nevertheless, **BO3d** exhibited intermolecular C–H⋯π interactions between the BO cores, with short distances of 2.722–2.845 Å (Supplementary Fig. 9), and C–H⋯π interactions between fluorenyl moieties in **BO4e** (Supplementary Fig. 12) and between Mes and BO core in **BO5** (Supplementary Fig. 13) were also observed. Most interestingly, **BO6** showed solid-state stacking with perpendicularly overlapping head–tail configurations with intermolecular C–H⋯π distances of 3.049–3.372 Å (Fig. 4, and Supplementary Fig. 14). These results reveal that both the BO cores and Mes groups considerably affected the solid-state structures. The ubiquitous π-stacks are beneficial for charge carrier transfer in organic electronic applications[6,28].

## Theoretical analyses

To further understand the aromaticity and electronic properties of the dBO-PAHs, we performed density functional theory (DFT) and nucleus-independent chemical shift (NICS) calculations (Fig. 3, and Supplementary Figs. 15–19). The BO-fused rings typically had NICS(1) values from −2.1 to −2.8, which were comparable to that of the previously reported BN-fused dibenzo[g,p]chrysene (−2.9)[16], suggesting a low aromaticity. Additionally, all the neighboring phenyl rings showed much more negative NICS(1) values, similar to their corresponding PAH analogs, e.g., **BO1a** (−11.58 and −11.57) *vs.* **CC1a** (−13.71 to −13.64) (Supplementary Fig. 15), revealing that the BO moieties hardly affected the aromaticity of the surrounding phenyl rings. Both the above effects enhance the stability of the dBO-PAHs.

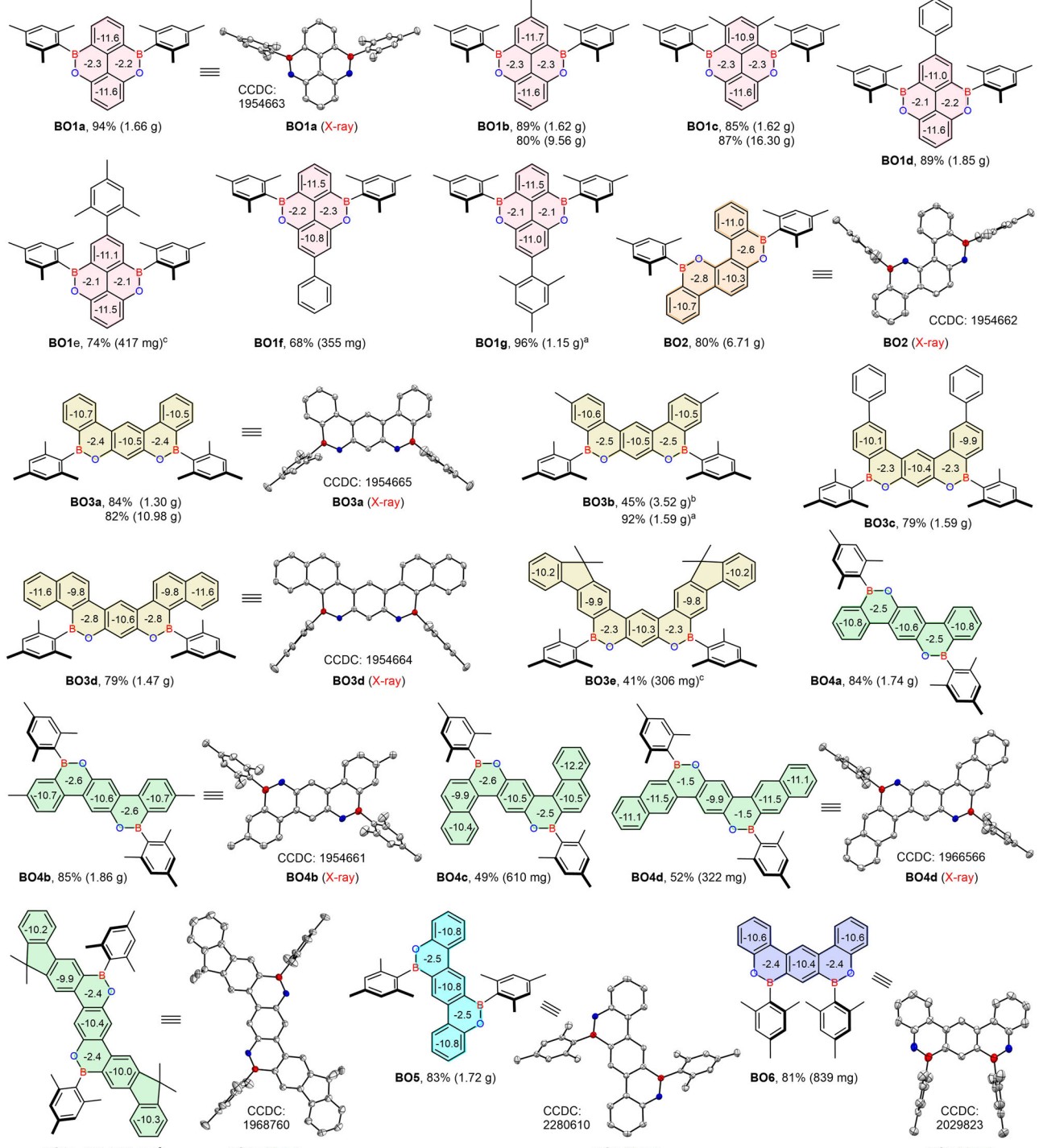

**Fig. 3 | Skeletal editing and theoretical calculation.** The dBO-PAHs synthesized via one-pot protocol with 4 or 8 mol% AlCl₃ was used. The single-crystal X-ray structures are shown with thermal ellipsoids at 50% probability, and hydrogen atoms are omitted for clarity. NICS(1) values calculated with a B3LYP/6-31 G(d) basis set in the gas phase are shown. The obtained product weight was provided in parentheses. ᵃ20 mol% AlCl₃ was used. ᵇA by-product 6-mesityl-8-methyl-2-(*p*-tolyl) −6*H*-dibenzo[*c*,*e*][1,2]oxaborinin-3-ol (**BO3b-OH**, CCDC: 2290997) was also isolated in 41% yield. ᶜ2.0 equiv AlCl₃ was used.

The DFT calculations showed that the dBO-PAHs had similar lowest unoccupied molecular orbital (LUMO) levels to their corresponding carbon analogs; however, they possessed remarkably stabilized highest occupied molecular orbitals (HOMOs), resulting in significantly large HOMO–LUMO energy gaps ($E_g$), this was attributed to the delocalization effect for BO1 series and **BO6**, and the electron-deficient property of the boron for **BO2**, BO3 series, BO4 series and **BO5** (Supplementary Figs. 16–19); for instance, the LUMO, HOMO, and $E_g$ values were −1.47, −5.28, and 3.81 eV for **CC1a** vs. −1.50, −6.06, and 4.56 eV for **BO1a**, respectively (Supplementary Table 5). As illustrated in Supplementary Figs. 16–19, the LUMOs of all dBO-PAHs except **BO4d** showed a nonuniform distribution on the whole BO cores, and were predominantly located on the boron atoms and the connected benzene rings. The HOMOs of **BO2, BO3a, BO4a**, and **BO5** fully distributed on the whole BO cores, whereas they were predominantly on the Mes groups for **BO1a** and **BO6**

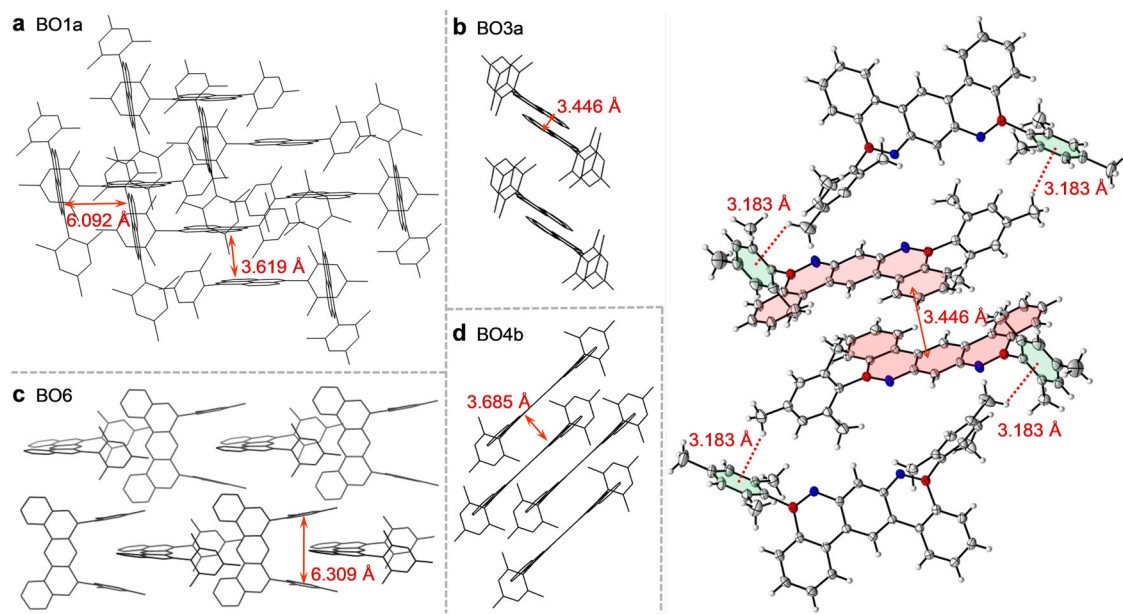

**Fig. 4 | Single crystal structures.** Crystal packing structures of (**a**) BO1a, (**b**) BO3a, (**c**) BO4b and (**d**) BO6. Double arrows represent the distance between two planes; dot lines represent the intermolecular C−H···π interactions.

(Supplementary Fig. 17). The fused phenyl groups (**BO3d, BO4c, BO4d**) and incorporated fluorenyls (**BO3e, BO4e**) significantly extended both the HOMO and LUMO distributions (Supplementary Figs. 18, 19). The substituent phenyl groups in **BO3c** extended the LUMO distribution, but hardly affected the HOMO distribution (Supplementary Fig. 18); however, the completely opposite effect was observed for **BO1d** (Supplementary Fig. 16). These results indicate that the skeletal editing, substituent modification, and tuning of the π-conjugation had great influence on the electronic structures of the dBO-PAHs, and could be also used to regulate their energy levels and gaps.

## Photophysical properties

The photophysical properties of the boron-containing PAHs are important for their applications in organic electronics. Thus, their absorption and emission properties were systematically investigated and illustrated in Fig. 5 and Supplementary Figs. 20–42, whereas the corresponding data were listed in Table 1. The strong absorption bands of the dBO-PAHs below 310 nm were assigned to π–π* transitions. The bands in the low-energy region, typically > 325 nm, were identified as intramolecular charge transfer (ICT) transitions introduced by the electron-deficient BO moieties. Introducing electron-donating group (Me) or extending the π-conjugation gradually red-shifted the highest-wavelength absorption band, e.g., 354, 357, 397, 376, and 385 nm for **BO4a, BO4b, BO4c, BO4d**, and **BO4e**, respectively, and the same effect was also observed for the BO3 series (Fig. 5, Supplementary Fig. 20). The fused π-conjugation from the C–C bond at the β-position of the naphthyl moiety dramatically increased the absorption intensities of both π–π* and ICT transitions, e.g., $\varepsilon = 5.57 \times 10^4$ (262 nm) and $3.47 \times 10^4$ (290 nm) for **BO3a** vs. $\varepsilon = 12.98 \times 10^4$ (284 nm) and $4.87 \times 10^4$ (370 nm) for **BO3d**, as well as $\varepsilon = 2.26 \times 10^4$ (307 nm) and $2.95 \times 10^4$ (354 nm) for **BO4a** vs. $\varepsilon = 8.52 \times 10^4$ (295 nm) and $5.21 \times 10^4$ (376 nm) for **BO4d** (Table 1, Fig. 5, Supplementary Fig. 20). Additionally, the experimentally calculated oscillator strength ($f$) of $S_1$-$S_0$ for **BO4e** was estimated to be 0.24 (see supporting information for details), the high $f$ was attributed to the $C_2$ symmetry of the molecular geometry, and extended linear conjugation system by two fluorenyl units, as well as the incorporation of the BO units, which could induce a larger transition dipole moment compared to no BO-fused PAHs[29,30].

Excitingly, all the dBO-PAHs showed strong emission in dichloromethane at room temperature, peaking at 355–414 nm with relatively small full width at half-maximum (FWHM) values of 25–57 nm (Table 1, Supplementary Figs. 21, 23–42). Most dBO-PAHs exhibited well-resolved vibronic emission spectra, and dBO-PAHs with large π-conjugation showed one dominant emission peak with small FWHM value, like **BO3d, BO3e** and **BO4e**; by contrast, **BO1f, BO1g** and **BO2** displayed unstructured and broad emission spectra, resulting in large FWHM values, this should be attributed to their relatively strong electronic vibronic coupling between the $S_0$ and excited states as well as by structural relaxation at the excited states. The dBO-PAHs also possessed $\Phi_{PL}$ values of 23–95%, with short excited state lifetimes ($\tau_f$) of 1.2–5.8 ns, resulting in large radiative rates ($k_r$) typically at the order of $10^8$ (Table 1). BO1 series, **BO2** and **BO6** had low quantum efficiencies due to their small $k_r$ and large nonradiative rate ($k_{nr}$); BO3, BO4 series and **BO5** showed high quantum efficiencies because of the enhanced radiative rate $k_r$. In particular, **BO4a** and **BO4b** had relatively low $\Phi_{PL}$ of 62% and 50%, respectively; **BO4e** showed an ultra high $\Phi_{PL}$ of 95%, together with a short $\tau_f$ of 1.4 ns, leading to a large $k_r$ of $6.8 \times 10^8 \, s^{-1}$ and a small $k_{nr}$ of only $0.4 \times 10^8 \, s^{-1}$ (Table 1). The high $\Phi_{PL}$ of **BO4e** was attributed to the ICT characteristics, reflecting the importance of the BO incorporation; and also **BO4e** had larger $f$ (0.24) than those of **BO4a** ($f = 0.092$) and **BO4b** ($f = 0.12$) (see supporting information for details). These results suggest the potential application of the present compounds as near-ultraviolet or deep-blue emitters in OLEDs, especially for the BO3 and BO4 series with high $\Phi_{PL}$ (Table 1).

## Ultralong low-temperature phosphorescence

Single-component, dual-emissive materials are highly desirable because of their potential applications in bio-imaging, ratiometric/optical sensing, and oxygen detection[31-33]. Ultralong organic phosphorescence (UOP) materials also have wide and promising applications in the areas of surface icing indications, optical thermometry, biological labeling, and virus preservation in extreme temperature conditions[34-37]. Traditionally, the UOP materials have been extensively developed via host-guest doping, crystallization, polymerization, etc.[38] however, single-component UOP materials with long second-level phosphorescence lifetimes ($\tau_p$) are rare[39-41]. At 77 K in 2-MeTHF, most of the dBO-PAHs, especially the BO1 series, **BO2, BO3a–BO3d,**

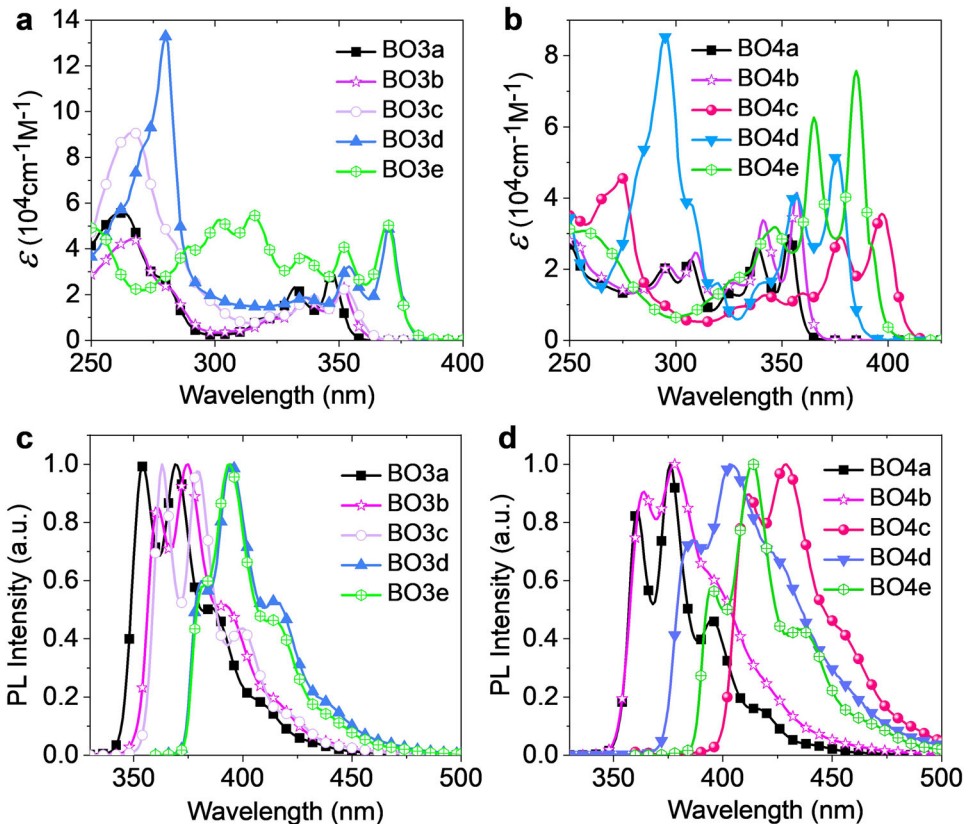

**Fig. 5 | Absorption and PL spectra.** Comparison of absorption and PL spectra of (**a**, **c**) BO3a, BO3b, BO3c, BO4d, and B4e, and (**b**, **d**) BO4 series in dichloromethane.

**BO4a, BO4b, BO5**, and **BO6**, exhibited prominent single-component dual emissions with nanosecond fluorescence ($\tau_f = 1.2$–12.6 ns) and UOP ($\tau_p = 0.3$–5.0 s) (Table 1, Fig. 6, Supplementary Figs. 23–44).

As shown in Fig. 6a, **BO1a** showed dual emission at 77 K, in which the 330–400 nm fluorescent component with $\tau_f$ 3.6 ns was in good agreement with the RT spectrum with dominant peaks at 348 and 362 nm and $\tau_f$ 2.4 ns; a much stronger 400–550 nm phosphorescent component could be selectively recorded by time-resolved PL spectroscopy, and was also in line with the steady-state PL spectrum at 77 K. Similar low-temperature phosphorescence spectra could be also recorded for **BO1d, BO1e, BO4a, BO5** and **BO6**, whose fluorescent components completely disappeared because of their short lifetimes (Fig. 6b–f). Additionally, the ratios of the fluorescence and phosphorescence intensities for the dBO-PAHs were strongly dependent on the nature of the double BO cores and the electronic effect of the substituents on the cores; taking the BO1 series as an example, the incorporation of electron-donating groups into the core or the extension of the conjugation system of the core resulted in a decreased phosphorescence intensity (Fig. 6, Supplementary Figs. 23–42). Importantly, the long second-level $\tau_p$ (Fig. 6g, Supplementary Fig. 44) enabled the dBO-PAHs to exhibit strong and ultralong low-temperature afterglow of up to 20 s after turning off the UV light source (Fig. 6a–f, Supplementary Figs. 45, 46). Moreover, the dBO-PAHs also exhibited distinct color changes for the low-temperature UOP after switching the UV light off, e.g., deep blue *vs.* greenish blue for **BO1d** and **BO1e**, deep blue *vs.* sky blue for **BO2, BO3b, BO3c** and **BO6**, deep blue *vs.* green for **BO4a** and **BO5**, and blue vs. yellow for **BO4c** and **BO4d** (Fig. 6a–f, Supplementary Figs. 45, 46); this highlighted their potential application in anti-counterfeiting and information encryption[40,42]. Notably, no UOP property was observed for the dBO-PAHs in PMMA films at room temperature or the ice point.

Typically, metal-free organic molecules lack a sufficient intersystem crossing (ISC) from their singlet to triplet states and thus

exhibit a rapid radiative decay to the ground states, resulting in no or very weak phosphorescence. Therefore, halogens or carbonyls need to be introduced into the phosphorescent organic molecules to facilitate the spin-orbital coupling (SOC) and further enhance the ISC rate;[38] however, these strategies enhance the rates of both $S_n \to T_n$ and $T_n \to S_0$ transitions, resulting in shorter phosphorescent lifetimes[41]. Thus, the design and development of low-temperature UOP is still a challenge, because this is required to accelerate the ISC from $S_1$ to triplet state and simultaneously decelerate the radiative decay from $T_1$ to ground state, achieving to accumulate enough triplet excitons. To obtain a deep understanding of the UOP properties of the dBO-PAHs, TD-DFT calculations were performed on their excited state levels and some main transition configurations (Fig. 6h, i, Supplementary Figs. 47–51). As illustrated in Fig. 6h, **BO1a** had an extremely small $S_1$-$T_8$ energy gap ($\Delta E_{S1-T8}$) of −0.023 eV, along with the same transition orbital composition of 97% HOMO → LUMO for $S_1$ and 73% HOMO → LUMO for $T_8$, facilitating an efficient ISC from $S_1$ to $T_8$; TD-DFT calculations also showed that SOC values of $\xi(S_1$-$T_7)$ and $\xi(S_1$-$T_5)$ were very large, 1.01 and 1.61 cm$^{-1}$, respectively, together with small $\Delta E_{ST}$ values of 0.185 and 0.052 eV, revealing that the ISC from $S_1$ to $T_7$ and $T_5$ were also efficient pathways according the Fermi's golden rule. Moreover, natural transition orbital (NTO) analyses indicated that ISC from $S_1$ to $T_8$ was complete local excited (LE) character, and ISC from $S_1$ to $T_7$ had hybridized local and charge transfer (HLCT) characters in the molecular core (Supplementary Fig. 47). Then, the internal conversion (IC) from $T_8$, $T_7$, $T_5$, to $T_1$ can occur rapidly, due to several small energy gaps separating them. In addition, the large $\Delta E_{S1-T1}$ of 0.386 eV and the mismatched transition orbital compositions of $T_1$ and $S_1$ prevent the reverse ISC from $T_1$ to $S_1$. Moreover, the rigid molecular geometry of **BO1a** (Supplementary Fig. 4) and the surrounding solid environment suppress the nonradiative decay. All the above factors endow **BO1a** with a long $\tau_p$ (3.1 s). Similarly, **BO6** exhibited an even smaller $\Delta E_{S1-T2}$ of only 0.013 eV, the same transition orbital composition for $S_1$ (99%

**Table 1 | Photophysical properties of dBO-PAHs**

| BO-PAH | absorption at RT[a] | emission at RT[a] | | | | | | emission at 77 K[b] | | | |
|---|---|---|---|---|---|---|---|---|---|---|---|
| | $\lambda_{abs}$ [nm] {ε × 10$^{-4}$ [cm$^{-1}$ M$^{-1}$]} | $\lambda_{PL}$ [nm] | FWHM [nm] | $\Phi_{PL}$ [%] | $\tau_f$ [ns][c] | $k_r$ [10$^8$ s$^{-1}$] | $k_{nr}$ [10$^8$ s$^{-1}$] | $\lambda_P$ [nm] | $\tau_f$ [ns][c] | $\tau_P$ [s][d] | $E_{T1}$ [eV] |
| BO1a | 264 (2.63), 324 (0.45), 338 (0.44) | 348 (sh),362 | 41 | 26 | 2.4 | 1.1 | 3.1 | 405 | 3.6 | 3.1 | 3.06 |
| BO1b | 266 (2.21), 330 (0.43), 344 (0.45) | 355,368 (sh) | 38 | 28 | 2.9 | 1.0 | 2.5 | 413 | 4.3 | 3.3 | 3.00 |
| BO1c | 254 (4.00), 325 (0.44), 338 (0.43) | 347 (sh),360 | 37 | 27 | 3.4 | 0.8 | 2.1 | 410 | 7.2 | 3.9 | 3.02 |
| BO1d | 281 (3.42), 337 (0.48), 350 (0.40) | 369,384 (sh) | 42 | 39 | 5.8 | 0.7 | 1.1 | 471 | 4.6 | 4.0 | 2.63 |
| BO1e | 284 (3.26), 335 (0.45), 348 (0.41) | 356 (sh),369 | 43 | 23 | 3.3 | 0.7 | 2.3 | 413 | 4.5 | 2.9 | 3.00 |
| BO1f | 290 (4.96), 348 (0.35) | 377 | 49 | 26 | 3.3 | 0.8 | 2.2 | 467 | 3.0 | 3.2 | 2.66 |
| BO1g | 268 (2.94), 329 (0.38), 341 (0.33) | 371 | 50 | 24 | 3.1 | 0.8 | 2.5 | 412 | 3.8 | 3.2 | 3.01 |
| BO2 | 266 (5.71), 316 (0.16) | 361 | 52 | 26 | 2.1 | 1.2 | 3.5 | 430 | 4.8 | 3.5 | 2.88 |
| BO3a | 262 (5.57), 332 (2.32), 347 (2.90) | 354,369 | 37 | 74 | 1.8 | 4.1 | 1.4 | 430 | 2.3 | 3.8 | 2.88 |
| BO3b | 267 (4.41), 336 (1.87), 351 (2.31) | 361 (sh),375 | 37 | 62 | 1.9 | 3.3 | 2.0 | 435 | 2.5 | 3.7 | 2.85 |
| BO3c | 267 (9.09), 338 (1.86), 353 (2.25) | 363,379 (sh) | 29 | 51 | 1.2 | 4.3 | 4.1 | 485 | 1.7 | 3.7 | 2.56 |
| BO3d | 284 (12.98), 354 (3.23), 370 (4.87) | 394 | 40 | 61 | 2.2 | 2.8 | 1.8 | 498 | 3.0 | 1.1 | 2.49 |
| BO3e | 315 (5.48), 352 (4.07), 369 (5.11) | 394 | 28 | 78 | 2.5 | 3.1 | 0.9 | 482 | 2.2 | 3.8 | 2.57 |
| BO4a | 307 (2.26), 338 (2.62), 354 (2.95) | 361 (sh),376 | 25 | 62 | 1.5 | 4.1 | 3.2 | 478 | 2.0 | 3.9 | 2.59 |
| BO4b | 309 (2.46), 341 (3.38), 357 (4.15) | 364 (sh),378 | 45 | 50 | 2.0 | 2.5 | 2.5 | 446 | 3.7 | 3.0 | 2.78 |
| BO4c | 360 (1.32), 377 (2.89), 397 (3.55) | 412 (sh),429 | 41 | 72 | 2.2 | 3.3 | 1.3 | 580 | 2.2 | 0.3 | 2.14 |
| BO4d | 339 (1.57), 357 (4.12), 376 (5.21) | 386,403 | 57 | 84 | 8.4 | 1.0 | 0.2 | 561 | 12.6 | 1.1 | 2.21 |
| BO4e | 347 (3.18), 365 (6.26), 385 (7.57) | 395 (sh),414 | 31 | 95 | 1.4 | 6.8 | 0.4 | 532 | 1.2 | 1.1 | 2.33 |
| BO5 | 264 (4.42), 303 (4.63), 355 (1.51) | 388 (sh),408 | 51 | 46 | 4.4 | 1.1 | 1.2 | 475 | 3.6 | 3.6 | 2.61 |
| BO6 | 274 (4.52), 307 (2.98), 350 (1.12) | 354 (sh),369 | 36 | 27 | – | – | – | 441 | 2.4 | 5.0 | 2.81 |

$\lambda_{abs}$ absorption wavelength, ε molar extinction coefficient, $\Phi_{PL}$ photoluminescence efficiency, $\tau_f$ fluorescence lifetime, $k_r$ radiation rate, $k_{nr}$ non-radiation rate, $\lambda_P$ phosphorescence lifetime, $E_{T1}$ the energy level of the lowest triplet excited state.

[a]Measured in dichloromethane.
[b]Measured in 2-MeTHF.
[c]Excited state lifetime of fluorescence.
[d]Excited state lifetime of phosphorescence. $k_r = \Phi_{PL}/\tau$; $k_{nr} = (1-\Phi_{PL})/\tau$; $E_{T1} = 1240/\lambda_P$.

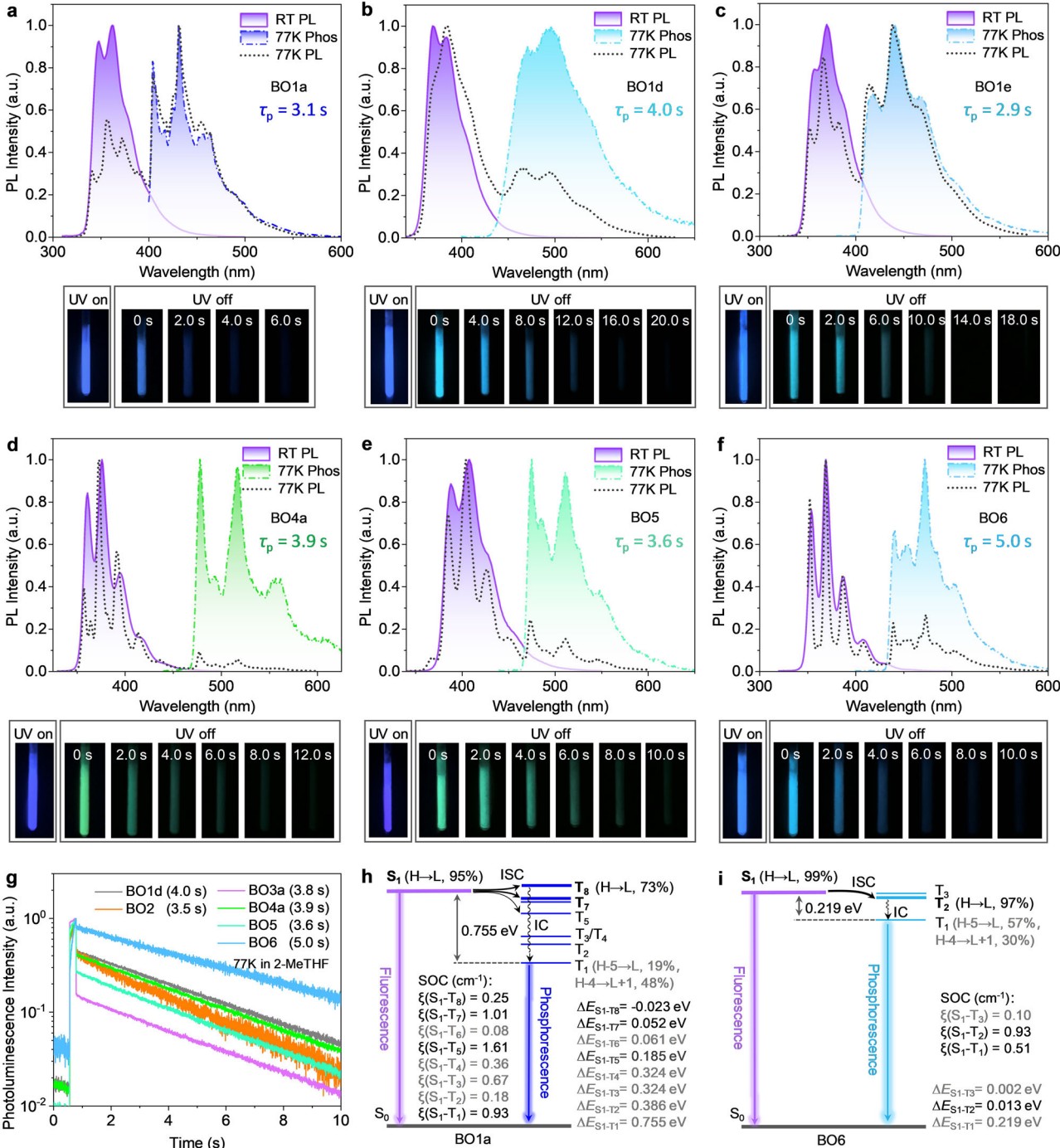

**Fig. 6 | Photophysical properties and state energy diagrams of dBO-PAHs.**
**a–f** Comparison of room-temperature PL spectra in dichloromethane, as well as 77 K PL and phosphorescent spectra in 2-MeTHF of BO1a, BO1d, BO1e, BO4a, BO5, and BO6, along with corresponding ultralong organic phosphorescence (UOP) photographs at 77 K in THF; the corresponding 77 K phosphorescent lifetime ($\tau_P$) is shown in the inset. **g** Comparison of PL decay curves of BO1d, BO2, BO3a, BO4a, BO5, and BO6 at 77 K in 2-MeTHF upon excitation at their corresponding first phosphorescent peak. **h, i** TD-DFT-calculated singlet and triplet energy levels, and

main transition configurations of BO1a and BO6 obtained at the B3LYP/6-31 G(d) level based on the optimized $S_0$ geometry; as well as spin-orbit coupling (SOC) matrix elements between $S_1$ and $T_n$ states of BO1a and BO6 evaluated using PySOC at the B3LYP/6-31 G(d,p) level. RT room temperature, PL photoluminescence spectrum, Phos phosphorescence spectrum, THF tetrahydrofuran, 2-MeTHF 2-methyltetrahydrofuran, ISC intersystem crossing, IC internal conversion, $\xi$ SOC value, $\Delta E$ energy gap, H HOMO, L LUMO.

HOMO → LUMO) and $T_2$ (97% HOMO → LUMO) and large $\xi(S_1\text{-}T_2)$ of 0.93 cm⁻¹ (Fig. 6i); the ISC from $S_1$ to $T_2$ also showed complete LE character in the molecular core (Supplementary Fig. 47). In contrast, **BO1d** had small $\Delta E_{S1\text{-}T7}$ of 0.021 eV, but the $\xi(S_1\text{-}T_7)$ was very tiny (0.24 cm⁻¹), this resulted in slow ISC and relatively long UOP (Supplementary Fig. 47). These results provide a new perspective for the

molecular design and development of metal- and halogen-free, single-component, dual-emission materials with UOP.

## Thermal stability
Besides the chemical stability discussed above, the thermal stability of the doubly BO-PAHs is also important for their applications in organic

electronics[43,44]. TGA measurements showed that **BO1b**, **BO2**, **BO4a**, **BO5**, and **BO6** exhibited decomposition temperatures at 5% weight loss ($\Delta T_{5\%}$) of 297, 343, 387, 339, and 335 °C, respectively (Supplementary Fig. 52). The good thermal stabilities can be attributed to the high dissociation energy of the B–O bond (536 kJ/mol)[45]. Additionally, no glass transition temperature was observed in the corresponding differential scanning calorimetry (DSC) measurements (Supplementary Fig. 52), indicating good morphological stability.

## Performance of dBO-PAHs as host materials for high-brightness deep-blue OLEDs

OLEDs are advanced devices with applications in full-color displays and solid-state lighting; efficient and high-brightness deep-blue phosphorescent OLEDs (POLEDs) with Commission International de l'Eclairage (CIE) $y$ coordinate (CIE$_y$) below 0.20 are essential for outdoor application[46]. However, deep-blue POLEDs typically exhibit a severe efficiency decrease at high luminescence[47–49], known as efficiency roll-off. Because of the high $E_{T1}$ of the deep-blue phosphorescent emitters (2.70–2.75 eV), very few host materials can be selected to solve this problem; therefore, novel host materials with high $E_{T1}$ (above 2.80 eV) are highly needed[50]. As shown in Fig. 7a, compared to **CC1b**, BO incorporation greatly affected the electronic structure of **BO1b**, which exhibited a delocalized HOMO distribution over the whole molecule (that is, not only on the BO cores, but also on the Mes groups), resulting in a stabilized HOMO level and a larger $E_g$. Similar results were also observed for **BO1e** and **BO1g** (Supplementary Fig. 16, Supplementary Tables 5, 6). Additionally, **BO1b** exhibited a perfect reversible reduction process in the cyclic voltammetry (CV) measurement (Supplementary Fig. 53). Moreover, because of the weakened π-conjugation of the BO core, **BO1b** had a significantly higher calculated $E_{T1}$ (3.10 eV) than **CC1b** (2.11 eV), which was also in good agreement with the experimental value of 3.00 eV (Table 1). The large $E_g$, high $E_{T1}$, together with the high thermal stability of d**BO-PAHs** enable them to serve as host materials for deep-blue OLEDs (Fig. 7b).

**BO1b** was investigated first. To demonstrate the potential of **BO1b** as host material, single carrier devices were fabricated (Supplementary Fig. 54); **BO1b** possessed a good electron-transporting ability, but hardly transported holes, suggesting that **BO1b** could serve as $n$-type host material. Then, deep-blue OLEDs were fabricated by vacuum deposition with a device architecture of ITO/HATCN (10 nm)/BPBPA (70 nm)/26mCDTPy (5 nm)/emitting layer (EML, 25 nm)/DPEPO (10 nm)/TmPyPB (30 nm)/Li$_2$CO$_3$ (1 nm)/Al, where the EML consisted of 8% PtON1:92% 26mCPy (device 1), 8% PtON1:92% mCP (device 2), 8% PtON1:92% mCBP (device 3), 8% PtON1:92% **BO1b** (device 4); PtON1 ($E_{T1}$ = 2.82 eV)[48] was used as blue emitter, and the traditional bipolar host 26mCPy ($E_{T1}$ = 2.92 eV)[46,51–54], $p$-type hosts mCP ($E_{T1}$ = 2.95 eV)[54] and mCBP ($E_{T1}$ = 2.95 eV)[50] were employed to compare the corresponding device performances (Fig. 7c, Supplementary Fig. 55, Table 2). All the four devices showed similar turn-on voltage, however, PtON1:**BO1b** device 4 exhibited a significantly narrow electroluminescence (EL) spectrum with FWHM of 50 nm, resulting in remarkable color purity improvement, and the CIE$_y$ value of device 4 (0.134) was much smaller than those of devices 1–3 (0.159–0.171) (Fig. 7d, e, Table 2). Importantly, compared with devices 1–3, device 4 also demonstrated enhanced peak external quantum efficiency (EQE) of 22.8%, which still retained high EQEs of 21.5% at 100 cd/m², and 16.3% at 1000 cd/m² (Fig. 7f, Table 2); but device 4 showed large efficiency roll-off in high brightness region of > 1000 cd/m² due to unbalanced exciton recombination. The EL spectra of the devices were slightly broader than that of the previous report[47], this could be attributed to the self-accumulation of the dopant because of differences in concentration, and self-absorption of other layers in the devices, such as BPBPA, on the basis of the narrowed spectra of the PtON1-doped thin films (Supplementary Fig. 56).

To solve the shortcoming and further improve the device performance, co-host systems of **BO1b** together with bipolar host 26mCPy (device 5), $p$-tpye hosts mCP (device 6) or mCBP (device 7) were employed to improve the exciton balance. As expected, devices 5, 6, and 7 demonstrated enhanced color purities, improved $L_{max}$s, and also significantly increased EQEs with dramatically reduced efficiency roll-offs compared to their corresponding single host devices 1–4 (Fig. 7g–i, Supplementary Fig. 57, Table 2). In particular, **BO1b**:mCBP-based device 7 achieved a peak EQE of 27.1%, and retained high EQEs of 25.2%, 21.8%, and 15.6% at 100, 1000 and 5000 cd/m², respectively, which was significantly better than the previous report with the same emitter[47] (Table 2). Surprisingly, **BO1b**:mCBP-based device 7 also achieved an $L_{max}$ value of up to 15722 cd/m², which was a 1.34-fold enhancement compared to mCBP-based device 3 and, and about 2-fold for **BO1b**-based device 4 in the same device settings (Table 2). This was attributed the improved charge balance in the EML and the reduction of charge accumulation at the interface between the EBL/EML or EML/HBL.

Brightness is also an important parameter of device performance, and high brightness is an urgent demand for the outdoor applications of display in smartphone. Our previously reported blue OLED employing PtON7-dtb as emitter demonstrated a high color purity and a high peak EQE, but suffered from the issue of large efficiency roll-off and small $L_{max}$ (-1555 cd/m²)[48]. Narrow-band blue emitter PtON7-dtb[48]-based device 8 with co-host of **BO1b**:mCBP was also fabricated (Fig. 7j–l, Table 2). Device 8 exhibited a narrow EL spectrum peaking at 453 nm with a FWHM of 28 nm and a CIE$_y$ of 0.088, and also realized a peak EQE of 27.6% with small efficiency roll-off, and retained high EQEs of 20.0%, 16.0% and 7.9% at 100, 1000 and 5000 cd/m², respectively; device 8 also achieved an $L_{max}$ of 5670 cd/m². Device 8 showed larger efficiency roll-off than that of device 7 with the same device structure due to the slightly higher $E_{T1}$ and longer $\tau$ of PtON7-dtb compared to PtON1[55]. PtON-TBBI-based deep-blue OLED demonstrated exceptionally long operational lifetime, but suffered from low color purity (CIE$_y$ = 0.197) and small $L_{max}$ (-2000 cd/m²)[46]. Employing PtON-TBBI ($E_{T1}$ = 2.78 eV, Supplementary Fig. 58) with short $\tau$ of 2.01 µs as emitter[46], device 9 demonstrated a peak EQE of 28.0%, which also represented the record-high EQE among Pt(II)-based deep-blue OLEDs with CIE$_y$ < 0.20[46–49,56–64], and was also among the highest EQE in the reported Ir(III)-based deep blue OLEDs[65–70] (Table 2, Supplementary Figs. 59, 60, Supplementary Tables 7, 8). Device 9 also showed narrow EL spectrum with a small FWHM of 21 nm, resulting in a huge color purity (CIE$_y$ = 0.104); moreover, a $L_{max}$ of 9377 cd/m² was also achieved (Table 2).

Then, other dBO-PAHs with high $E_T$ as hosts in deep-blue OLEDs were also investigated (Fig. 8, Supplementary Fig. 55, Table 2). The LUMO and HOMO levels of the dBO-PAHs were first calculated through their reduction potentials and photophysical properties (Supplementary Fig. 61, Supplementary Table 6). **BO1c** ($E_{T1}$ = 3.02 eV)-based device 10 and **BO1g** ($E_{T1}$ = 3.01 eV)-based device 11 using PtON1 as emitter also demonstrated good device performances with peak EQEs of 27.8% and 25.1%, retained high EQEs of 20.0% and 19.2% at 1000 cd/m², respectively; and they also achieved high $L_{max}$s of 27219 and 22683 cd/m², respectively; the $L_{max}$s were greatly increased compared to device 7 (15722 cd/m²). Importantly, the $L_{max}$ (27219 cd/m²) of device 10 represented the record-high $L_{max}$ among Pt(II), Ir(III) and BN-PAHs-based deep-blue OLEDs with CIE$_y$ < 0.20 (Fig. 7b, Supplementary Figs. 59, 60, 62, 63, Supplementary Tables 7–9)[46–49,56–64]. By comparison, PtON1:**BO2** device 12 and PtON1:**BO3a** device 13 exhibited significantly low EQEs and $L_{max}$s; this was attributed to the similar $E_{T1}$ of **BO2** (2.88 eV) and **BO3a** (2.88 eV) compared to PtON1 (2.82 eV)[48], which could not enable efficient energy transfer from host to emitter molecules. However, **BO2** and **BO3a**, as well as **BO6** ($E_{T1}$ = 2.81 eV) could be acted as hosts for PtON-TBBI ($E_{T1}$ = 2.78 eV)-based deep-blue OLEDs, devices 14, 15 and 16 showed narrow EL spectra with FWHM values of 21–22 nm, they also realized peak EQEs of 22.2%, 19.6% and 21.7%, and high $L_{max}$s of 14481,

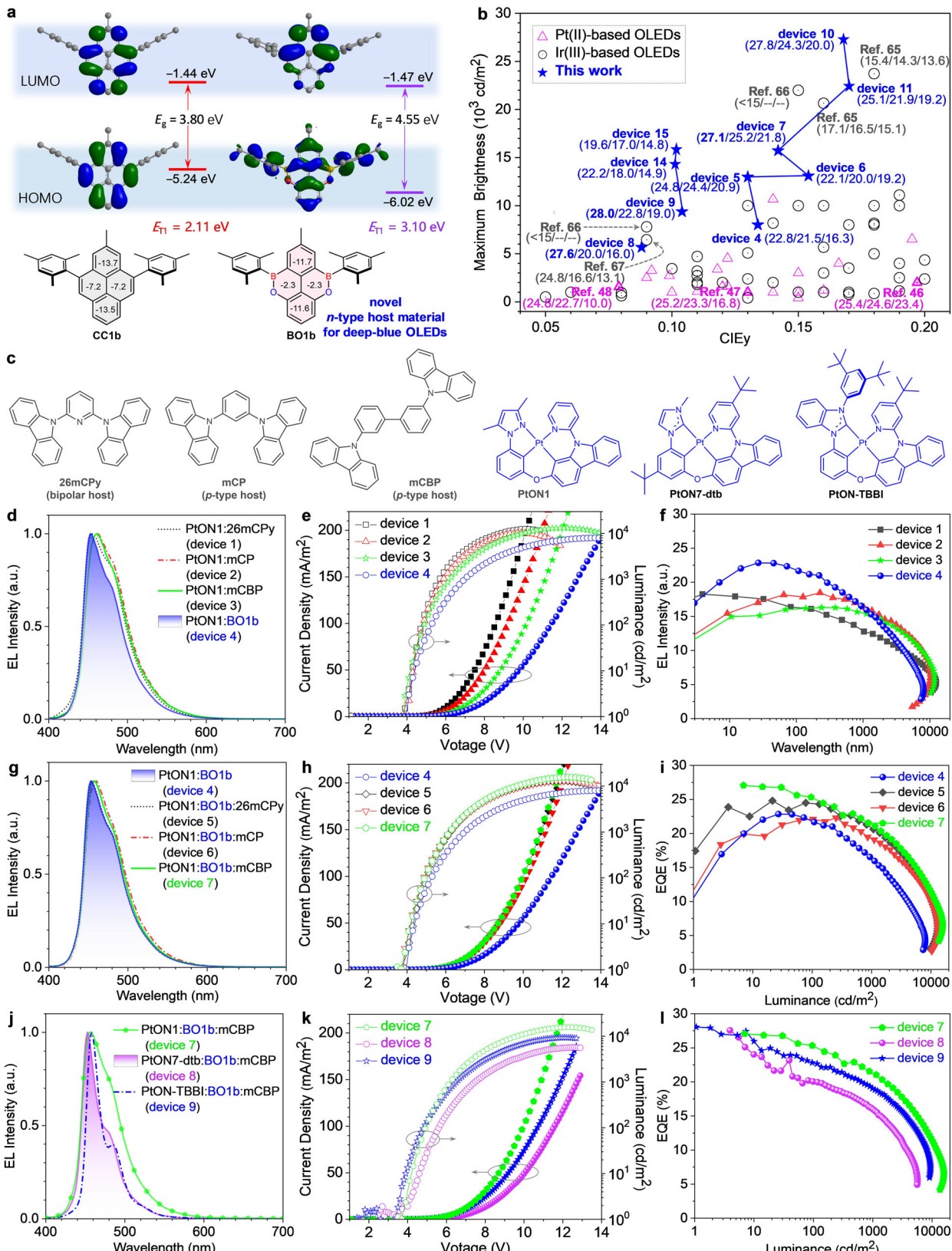

**Fig. 7 | Theoretical calculation, chemical structures and EL properties of deep-blue OLEDs. a** Comparison of calculated frontier orbital distributions, energy levels/gaps, and NICS(1) values of BO1b and its carbon analog CC1b. **b** CIE$_y$ vs. maximum brightness for Pt(II) and Ir(III)-based deep-blue OLEDs with CIE$_y$ < 0.20; devices connected with broken line to represent the devices with the same blue emitter; maximum EQE (%), and EQEs (%) at 100 and 1000 cd/m$^2$ were provided in parentheses for representative deep-blue OLEDs. see Table S7 and Table S8 for detailed information. **c**, Chemical structures of host materials and Pt(II)-based deep-blue emitters used in this study. **d**, **g**, **j** EL spectra of deep-blue OLEDs at 1000 cd/m$^2$. **e**, **h**, **k** Current density−voltage−luminance plots. **f**, **i**, **l** EQE $vs.$ luminance plots. $E_g$ the energy gap between HOMO and LUMO, $E_{T1}$ the energy level of the lowest triplet excited state, EL electroluminescence spectrum, EQE external quantum efficiency.

**Table 2 | Summary of device performance for Pt(II)-based deep-blue OLEDs and the comparison with representatively reported literatures**

| Device | EML | Voltage[a] [V] | $\lambda_{EL}$ [nm] | FWHM [nm] | CIE(x, y) | EQE[b] [%] | $L_{max}$ [cd/m²] |
|---|---|---|---|---|---|---|---|
| device 1 | 8% PtON1:92% 26mCPy | 3.8/5.6 | 454 | 57 | (0.143, 0.159) | 18.3/17.9/13.7/9.6 | 15271 |
| device 2 | 8% PtON1:92% mCP | 4.0/5.8 | 462 | 56 | (0.138, 0.171) | 18.5/17.9/16.3/12.0 | 10284 |
| device 3 | 8% PtON1:92% mCBP | 3.8/6.2 | 461 | 55 | (0.139, 0.167) | 16.3/16.0/15.4/11.8 | 11740 |
| device 4 | 8% PtON1:92% **BO1b** | 4.0/6.9 | 453 | 50 | (0.138, 0.134) | 22.8/21.5/16.3/8.8 | 8007 |
| device 5 | 8% PtON1:46% **BO1b**:46% 26mCPy | 3.9/6.2 | 454 | 50 | (0.138, 0.130) | 24.8/24.4/20.9/15.0 | 12982 |
| device 6 | 8% PtON1:46% **BO1b**:46% mCP | 3.8/6.0 | 460 | 52 | (0.138, 0.154) | 22.1/20.0/19.2/14.1 | 13098 |
| device 7 | 8% PtON1:46% **BO1b**:46% mCBP | 3.5/6.2 | 456 | 51 | (0.138, 0.142) | 27.1/25.2/21.8/15.6 | 15722 |
| device 8 | 8% PtON7-dtb:46% **BO1b**:46% mCBP | 3.4/7.4 | 453 | 28 | (0.138, 0.088) | 27.6/20.0/16.0/7.9 | 5670 |
| device 9 | 8% PtON-TBBI:46% **BO1b**:46% mCBP | 3.4/6.5 | 458 | 21 | (0.134, 0.104) | 28.0/22.8/19.0/13.4 | 9377 |
| device 10 | 8% PtON1:46% **BO1c**:46% mCBP | 3.1/5.5 | 456 | 56 | (0.141, 0.168) | 27.8/24.3/20.0/14.4 | 27219 |
| device 11 | 8% PtON1:46% **BO1g**:46% mCBP | 3.8/5.9 | 458 | 56 | (0.140, 0.170) | 25.1/21.9/19.2/14.7 | 22683 |
| device 12 | 8% PtON1:46% **BO2**:46% mCBP | 4.4/7.6 | 462 | 64 | (0.146, 0.208) | 17.3/17.2/14.5/9.2 | 9867 |
| device 13 | 8% PtON1:46% **BO3a**:46% mCBP | 4.9/7.9 | 462 | 61 | (0.143, 0.194) | 15.6/15.3/14.1/9.6 | 9803 |
| Device 14 | 8% PtON-TBBI:46% **BO2**:46% mCBP | 3.2/5.8 | 458 | 21 | (0.135, 0.103) | 22.2/18.0/14.9/10.4 | 14481 |
| Device 15 | 8% PtON-TBBI:46% **BO3a**:46% mCBP | 3.8/6.1 | 459 | 22 | (0.134, 0.107) | 19.6/17.0/14.8/10.8 | 15765 |
| Device 16 | 8% PtON-TBBI:46% **BO6**:46% mCBP | 3.9/6.7 | 459 | 22 | (0.133, 0.111) | 21.7/16.2/13.4/8.7 | 10071 |
| Ref. 46 | 13% PtON-BBI:27% SiTrzCz2:60% SiCzCz | 2.8/6.3 | 455 | 48 | (0.141, 0.197) | 25.4/24.6/23.4/--- | 2000 |
| Ref. 47 | 6% PtON1: 94% 26mCPy | 3.9/6.3 | 454 | 47 | (0.15, 0.13) | 25.2/23.3/16.8/--- | 1000 |
| Ref. 48 | 6% PtON7-dtb:47% PO15:47% TAPC | 2.8/4.4 | 451 | 29 | (0.148, 0.079) | 24.8/22.7/11.0/--- | 1555 |

$\lambda_{EL}$ electroluminescence wavelength, $L_{max}$ maximum brightness.
[a]Votage at 1 and 1000 cd/m².
[b]Maximum EQE, and EQEs at 100, 1000 and 5000 cd/m².

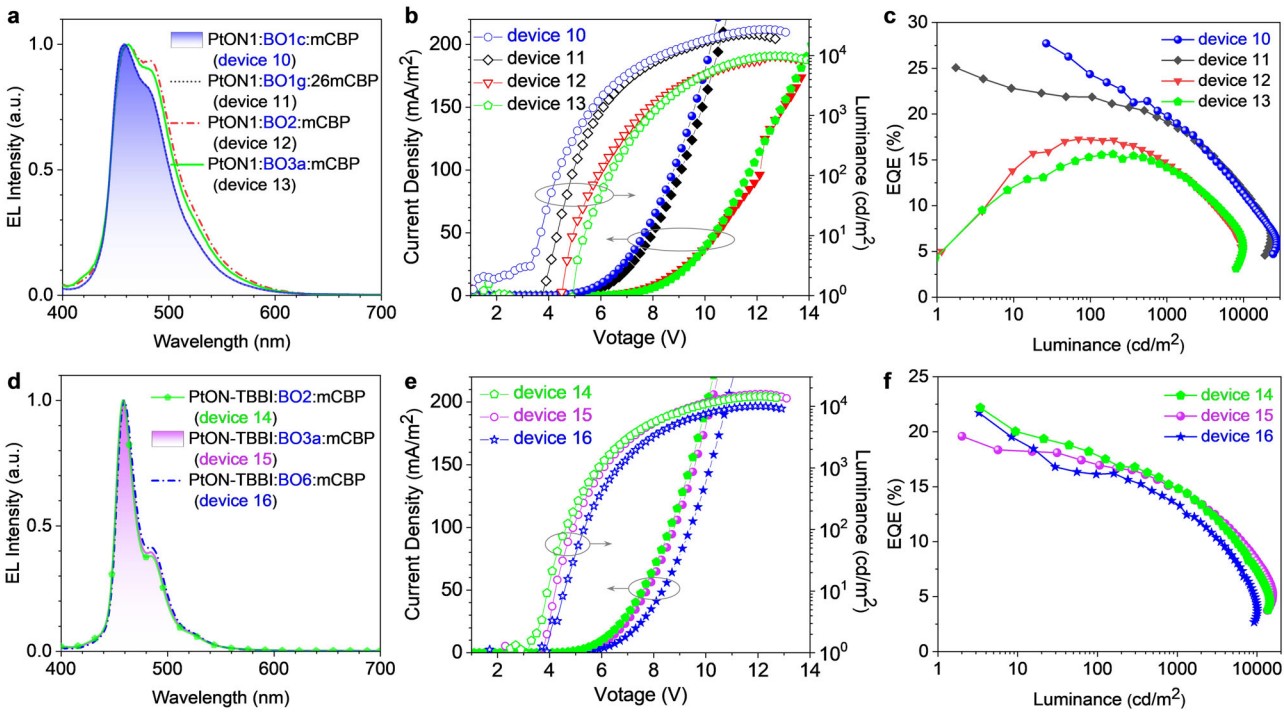

**Fig. 8 | EL properties of deep-blue OLEDs. a, d** EL spectra of deep-blue OLEDs at 1000 cd/m². **b, e** Current density−voltage−luminance plots. **c, f** EQE *vs.* luminance plots. EL, electroluminescence spectrum; EQE, external quantum efficiency.

15765 and 10071 cd/m², respectively. Last, the operational lifetimes of some deep-blue OLEDs were measured (Supplementary Fig. 64); because of the poor stability of DPEPO, they showed LT$_{50}$ with $L_0 = 500$ cd/m² of 0.2–1.0 h. However, PtON1:**BO1c**:mCBP device 10 and PtON1:**BO1g**:mCBP device 11 exhibited obviously longer operational lifetimes than those of PtON1:26mCPy device 1, this revealed that the

co-hosts-based OLEDs with improved charge balance increased the operational lifetime. Moreover, although PtON1 ($E_{T1} = 2.82$ eV)[48] had higher $E_{T1}$ than that of PtON-TBBI ($E_{T1} = 2.78$ eV), PtON1-based devices 10 and 11 also demonstrated significantly longer operational lifetimes compared to PtON-TBBI-based devices 14, 15 and 16, which revealing that the CIE$_y$ value also had a great effect on the operational lifetime.

The device performance should be further improved by optimizing the ratio of $n$-type dBO-PAHs and $p$-type host in the EML and employing advanced functional materials of other layers. The above studies demonstrate that **BO1b** and other dBO-PAHs with high $E_{T1}$, e.g., **BO1a, BO1c, BO1e, BO1g, BO2, BO3a, BO3b** and **BO6** (Table 1), can act as $n$-tpye host materials for high-brightness deep-blue OLEDs. This study provides a design concept for the development of host materials for deep-blue OLEDs, and also broadens the application of the dBO-PAHs in the field of organic electronics.

## Discussion

In this work, we developed a facile synthesis of dBO-PAHs through an efficient one-pot strategy consisting of tandem demethylation-electrophilic borylation-nucleophilic substitution reactions. Using this method, six types of dBO-PAHs were successfully synthesized with high regioselectivity and efficient skeletal editing. The dBO-PAHs showed high chemical and thermal stabilities. Single crystal X-ray diffraction analyses suggested that the enforced planarity of the BO cores and Mes groups effectively enhanced their chemical stability, and also greatly affected their solid-state stacking. Theoretical calculations revealed that the BO-fused rings showed low aromaticity and could effectively regulate the electronic properties of the dBO-PAHs. The dBO-PAHs exhibited strong fluorescence emission at room temperature in dichloromethane and achieved a $\Phi_{PL}$ and $k_r$ values of up to 95% and $6.8 \times 10^8\,\text{s}^{-1}$, respectively. In contrast, at 77 K in 2-MeTHF, most dBO-PAHs exhibited single-component, dual-emissions with nanosecond fluorescence (1.2–12.6 ns) and second-level UOP (0.3–5.0 s); importantly, the long $\tau_p$ enabled the dBO-PAHs to exhibit low-temperature ultralong afterglow of up to 20 s after UV light turn-off. Moreover, the BO incorporations endowed the dBO-PAHs to possess enlarged $E_g$ and an increased $E_{T1}$, making them ideal $n$-type host materials for high-brightness and high-efficiency deep-blue OLEDs; compared to single host device, deep-blue OLEDs employing **BO1b**:mCBP as co-hosts demonstrated significantly enhanced $L_{max}$s, EQEs and reduced efficiency roll-offs; PtON1-based device 10 achieved the record-high $L_{max}$ (27219 cd/m$^2$) and PtON-TBBI-based device 9 realized the record-high EQE (28.0%) among Pt(II)-based deep-blue OLEDs with CIE$_y$ < 0.20. These results demonstrate the great potential of the dBO-PAHs as low-temperature ultralong afterglow and $n$-type host materials in optoelectronic applications, and can also help to understand the electronic and optical properties of the dBO-PAHs.

## Methods
### Synthesis and structure characterization
Unless noted, all commercial reagents were purchased and used as received without further purification. Toluene and $n$-hexane were dried with CaH$_2$ reflux, and distillation before use. $^1$H NMR spectra were recorded at 400 or 500 MHz Bruker "AVANCE III" nuclear magnetic resonance system in CDCl$_3$ or DMSO-$d_6$ solutions and chemical shifts were referenced to tetramethylsilane (TMS) or residual protiated solvent. If CDCl$_3$ was used as solvent, $^1$H and $^{13}$C NMR spectra were recorded with TMS ($\delta$ = 0.00 ppm) and CDCl$_3$ ($\delta$ = 77.00 ppm) as internal references, respectively. If DMSO-$d_6$ was used as solvent, $^1$H and $^{13}$C NMR spectra were recorded with TMS ($\delta$ = 0.00 ppm) and DMSO-$d_6$ ($\delta$ = 39.52 ppm) as internal references, respectively. All of the new compounds were analyzed for HRMS on a Waters mass spectrometer using electrospray ionization in the positive ion mode of ESI-Q-TOF.

### General experimental procedure for the synthesis of dBO-PAHs
The dimethoxy compound was added to a dry three-necked flask equipped with a magnetic stir bar and a condensor. The flask was then evacuated and backfilled with nitrogen, this evacuation and backfill procedure was repeated twice. Then solvent toluene and $n$-hexane were added under a nitrogen atmosphere, then BBr$_3$ was added

dropwise. The mixture was stirred at room temperature for 18–40 h monitoring by TLC until the demethylation was completely. Then AlCl$_3$ was added quickly (The AlCl$_3$ is highly hygroscopic and should be weighed quickly!), the flask was placed in oil bath (75 $^\circ$C) and stirred for 8–10 h, and then the mixture was cooled down to room temperature. Mesitylmagnesium bromide (MesMgBr) was then added dropwise, and the mixture was then stirred at room temperature for another 1–10 hours monitoring by TLC until the reaction was completely. The reaction mixture was concentrated under reduced pressure, and the residue was purified through column chromatography on silica gel using petroleum ether/dichloromethane as eluent to afford the desired dBO-PAH compound. Please see the Supporting Information for the synthetic details of the dBO-PAHs.

### X-ray crystallography
X-ray diffraction data were collected at 170 K on a Bruker D8 Venture diffractometer using graphite-monochromated Mo-K$\alpha$ radiation ($\lambda$ = 0.71073 Å) from a rotating anode generator. The crystal structures were solved by ShelXT or ShelXS and refined with full-matrix least-squares methods with anisotropic thermal parameters for all non-hydrogen atoms on F2 using SHELXL-2015. Hydrogen atoms were found from different Fourier maps but placed in calculated positions and were refined isotropically using a riding model[71–73]. The crystal data as well as the details of data collection and refinement are summarized in Supplementary Tables 1–4.

### Thermal properties
Thermogravimetric analysis (TGA) and differential scanning calorimetry (DSC) were performed by TA-Q50. The TGA curve was measured at a heating rate of 10 $^\circ$C/min from RT to 500 $^\circ$C under nitrogen flow, after eliminating residual thermal history of compound.

### Quantum chemical calculations
The theoretical calculations were performed using Gaussian 09. The molecular geometries of ground states (S$_0$) were optimized with the density functional theory (DFT) method. The DFT calculations were performed using a B3LYP function with a basis set of 6-31 G(d) for C, H, O, and B atoms. The spin-orbit coupling (SOC) matrix elements between S$_1$ and T$_n$ states of **BO1a, BO1d,** and **BO6** were evaluated using PySOC at the B3LYP/6-31 G(d,p) level[74].

### Electrochemistry
Cyclic voltammetry and different pulsed voltammetry were performed using a CH1760E electrochemical analyzer according previous report. 0.1 M tetra-$n$-butylammonium hexafluorophosphate was used as the supporting electrolyte, anhydrous $N,N$-dimethylformamide, was used as the solvents for the $E_{ox}$ and $E_{red}$ measurements, and the solutions were bubbled with nitrogen for 15 min prior to the test. Silver wire, platinum wire and glassy carbon were used as pseudoreference electrode, counter electrode, and working electrode, respectively. The scan rate was 300 mV/s. The redox potentials are based on the values measured from different pulsed voltammetry and are reported relative to an internal reference ferrocenium/ferrocene (Cp$_2$Fe/Cp$_2$Fe$^+$). The reversibility of reduction or oxidation was determined using CV.

### Photophysical measurements
The absorption spectra were measured on an Agilent 8453 UV–VS Spectrometer. Steady-state emission experiments and lifetime measurements were performed on a Horiba Jobin Yvon FluoroLog-3 spectrometer. Low temperature (77 K) emission spectra and lifetimes were measured in 2-MeTHF cooled with liquid nitrogen.

### OLED fabrication and characterization
All devices were fabricated by vacuum thermal evaporation, and were tested outside glove box after encapsulation. Prior to deposition, the

prepatterned ITO-coated glass substrates were cleaned by subsequent sonication in deionized water, acetone, and isopropanol. The metal layer and organic layers were fabricated by vacuum thermal evaporation on the cleaned indium-tin-oxide (ITO) glass substrate under vacuum ($<4 \times 10^{-4}$ Pa) with 4 Å/s deposition rate for aluminum cathode and 2 Å/s for organic layers. The device areas were 9 mm$^2$ (3 mm × 3 mm). The current density–voltage–luminance characteristics of OLEDs were measured using a Keithey 2400 Source meter and a Keithey 2000 Source multimeter equipped with a calibrated silicon photodiode. The electroluminescence (EL) spectra were recorded with a multichannel spectrometer (PMA12, Hamamatsu Photonics).

## Data availability

The authors declare that the main data supporting the findings of this study are available within the article and its Supplementary Information files. Extra data are available from the corresponding author upon request. Crystallographic data for the structures reported in this Article have been deposited at the Cambridge Crystallographic Data Centre, under deposition numbers CCDC 1954663 (**BO1a**), 1954662 (**BO2**), 1954665 (**BO3a**), 2290997 (**BO3b-OH**), 1954664 (**BO3d**), 1954661 (**BO4b**), 1966566 (**BO4d**), 1968760 (**BO4e**), 2280610 (**BO5**) and 2029823 (**BO6**). Copies of the data can be obtained free of charge via https://www.ccdc.cam.ac.uk/structures/. The new crystallographic structures of target molecules are also available within the accompanied files. Source data are provided with this paper [data https://doi.org/10.6084/m9.figshare.24146901][75] Source data are provided with this paper.

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

## Acknowledgements
This work was supported by the National Natural Science Foundation of China (Grant No. 22178319, G.L.; 21878276, G.L. and 22138011, Y.-B.S.). We thank Prof. Qisheng Zhang and Dr. Shaohai Chen's helpful discussion. We also thank Jiyong Liu from Zhejiang University for the help of the single crystal measurements and analyses and Zhejiang Hongwu Technology Co., Ltd for the device fabrications and measurements.

## Author contributions
G.L. and Y.-B.S. initiated and supervised the project. G.L. designed the materials and analyzed the data. K.X., J.Z., Q. Chu, J.D. Q. Chen, and Y.Y. synthesized and characterized the doubly BO-embedded emitters and hosts. K.X., J.Z., J.D., Q. Chen, and Y.Y. performed the photophysical and electrochemical measurements. X.F., K.X., and Y.-F.Y. performed the computational calculation. W.L. fabricated the OLEDs and measured the device performance. G.L. and K.X. contributed to the manuscript writing. All authors discussed the progress of the research and reviewed the manuscript.

## Competing interests
The authors declare no competing interests.
