## [Peer Review File · Nature Communications]

Double Boron–Oxygen-Fused Polycyclic Aromatic Hydrocarbons: Skeletal Editing and Applications as Organic Optoelectronic MaterialsREVIEWER COMMENTS

Reviewer #1 (Remarks to the Author):

In this manuscript, the authors synthesized and characterized a series of dBO-PAHs with low-temperature ultralong afterglow property, and also demonstrated the potential for OLEDs as n-type host materials. The results of low-temperature ultralong organic phosphorescence are interesting, but the explanation is unclear and is not suitable in its present form for publication in Nat. Commun.

Major issues

1. In page 5, "For the synthesis of central symmetric double BO-fused pyrenes (Type I), electron-rich alkyl and aryl substitutions on either of the phenyl rings were tolerated in moderate to excellent yields (53-94%), except for BO1g (35%)", and authors mentioned that "it is difficult to attain the second electrophilic borylation on an electron-deficient phenyl ring", after authors introduced somehow electron-donating groups such as methyl and phenyl group (BO1b, BO1c, and BO1d), the reaction yield keeps similar to BO1a around 90%, but in the case of BO1e, the reaction yield almost goes down to half (just 55%), furthermore, the reaction yield of BO1g is half lower than that of BO1a. So could you please explain the reasons for the lower reaction yield and show the byproducts during the reactions. Also the same question on the reactions of BO3a, BO3b and BO3c.
2. Some dBO-PAHs (like BO3d, BO3e and BO4e) exhibited well resolved vibronic emission spectra, but some dBO-PAHs (like BO1f, BO1g and BO2) exhibited unstructured and broad emission spectra, the authors should explain the differences of molecular vibrations.
3. The authors mentioned about the high PLQY (95%) of BO4e was attributed to the ICT characteristics, but other dBO-PAHs such as BO4a and BO4b also possess strong ICT characteristics from UV-Vis absorption spectra and showed much lower PLQY of 62% for BO4a and 50% for BO4b, respectively. The detailed explanation should be described.
4. In page 13, "All the above factors endow BO1a with a long tp (4.0 s)", but in the Fig.6a, tp of BO1a is 3.1 s, this makes me confused about the explanation why BO1a showed low-temperature UOP (6.0 s) but BO1d showed two times longer lifetime (20 s) than that of BO1a. The calculations of natural transition orbitals and spin-orbital coupling are necessary to discuss intersystem crossing process.
5. The rigid molecular structure of dBO-PAHs and the solid state at 77K low temperature suppress the nonradiative decay, and the neat film of dBO-PAHs is also a solid state, so how about the UOP property of dBO-PAHs in the film state at room temperature or the ice point?
6. dBO-PAHs based host material achieved high luminance such as device 7 (15722 cd/m²), the authors should explain the reason why this kind of host material can achieve such high luminance.

7. In Fig. 7 e, h, and k, the EQE-L plots showed zigzag-shape, this is possible to obtain a underestimated or overestimated results, the authors should recheck the device performances.

8. For blue phosphorescence emitters, OLED device stability is also very important, how about the device operation lifetime based on dBO-PAHs host materials.

9. Ref 32 in Table 2 is a wrong citation. I guess this is the correct citation "Sun, J., Ahn, H., Kang, S. et al. Exceptionally stable blue phosphorescent organic light-emitting diodes. Nat. Photon. 16, 212–218 (2022)"

Reviewer #2 (Remarks to the Author):

See attachment.

G. Li, Y. She et al. reported a series of double BO-doped PAHs by a one-pot borylation of OMe-substituted precursors. The skeletons could be edited by changing the connection of aryl groups and the positions of methoxyl groups. They also applied the dBO-PAHs as host materials for OLED device, which showed high efficiency. The authors have demonstrated potential borylation mechanism. The structures were well characterized, and the photophysical properties were well recorded and elucidated by theoretical studies. For these reasons, the reviewer thinks this work is quite interesting. In this context one also has to ask why the authors compared their EQE with Pt(II)-based compounds? The reported materials/molecules are organic materials, however, Pt(II)-based compounds are coordinated compounds. They do not belong to the same class of comparators. How about their EQE compared to the BN series of compounds (BN-PAHs or NBN-PAHs, such as: Adv.Optical Mater. 2022, 10, 2201714)? Is there any advantages compared with these BN-PAHs or NBN-PAHs for their BO-PAHs (such as the FWHM, Φ PL, CIE, etc.)? The newly synthesized compounds indisputably possess an obvious structural beauty and as such already the synthetic achievement itself is of considerable significance. Nevertheless, at the same time all these arguments in my opinion do not sufficiently justify publication of the manuscript in its present form in Nature Communications. Moreover, the manuscript suffers from several additional issues which need to be addressed prior to resubmission. To sum up, the quality of the manuscript is not yet sufficiently high for publication, and I recommend the manuscript to be rejected for now. After appropriate adjustments and the additional experiments as suggested below, it should be reconsidered as a new manuscript.

- (1) For the device part, why did the authors only apply **BO1b** for the device application, how about other **BO-PAHs**? For sure, compared to pure-carbon **CC1b**, **BO1b** has the advantages, but the authors did not give any explanation why they did not use other type of BO-PAHs in their manuscript.
- (2) 2D NMR spectra of **BO5** might be provided.
- (3) Type I, Type V and Type VI introduce two boron atoms to the same benzene rings. It seemed reasonable that Type II product was formed owing to the electronic effects. But the first borylation exclusively took place at the less steric methoxyl group?
- (4) The precursors with two methoxyl groups connected to the central benzene rings (*meta*-, or *para*-) could afford dBO-PAHs smoothly. Is it possible to further applied the one-pot strategy to the *ortho*-analog?

Reviewer #3 (Remarks to the Author):

In this manuscript, Li, She and coworkers developed a one-pot tandem demethylation-electrophilic substitution reactions for double-oxygen-fused polycyclic aromatic hydrocarbons (dBO-PAHs). Using this method, six types of dBO-PAHs have been successfully prepared and fully characterized by NMR, HRMS and X-ray analysis. Particularly, the incorporation of the B-O units endows dBO-PAHs with low-temperature ultralong afterglow of up to 20 s after UV excitation, which is a very interesting phenomenon that has received a lot of attention nowadays. Moreover, the potential applications of dBO-PAHs as host materials for metal complex-based deep-blue OLEDs have also been demonstrated. Despite a lot of works have been done, there are still some parts remained to be well clarified and the significance/inspiration of this work is not very great. Considering the high requirements of this journal, major revisions are needed after consideration.

More detailed comments are shown below:

1. The authors have claimed that the incorporation of B-O units enables the dBO-PAHs with unique optoelectronic properties, such as high fluorescence quantum yield in dichloromethane solution and single-component, low-temperature ultralong afterglow. However, the description on the structure-property logic is confusing. For example, the authors claimed that the low-aromaticity of BO-fused rings lead to high thermal and chemical stabilities (Page 2) as well as high quantum yields (Page 4) of dBO-PAHs. In fact, the good stabilities of dBO-PAHs are due to the synergistic effects of the p-pai conjugation, the delocalization of the lone electron pairs of the oxygen atoms as well as the bulky Mes moieties. Therefore, the authors are suggested to pay more attention to the logic flow.
2. Continuing the comment 2, comparing the optoelectronic properties of structurally related all carbon-based PAHs (such as CC1b) and BN-fused dBO-PAHs with those of dBO-PAHs in this work should be made.
3. On Page 5, the yield of BO1g is relatively low as compared with those of other central symmetric double BO-fused pyrenes (Type I). It seems like the relatively low yield is not due to the electronic effect or the steric effect, so the authors are suggested to give reasonable explanations on this.
4. On page 7, the almost planar BO-fused cores of dBO-PAHs are supported by the sum of degrees of one C-B-C and two C-B-O bonds, not three C-B-C bonds?
5. As revealed by DFT calculation results, dBO-PAHs had similar LUMO energy levels and remarkably low-lying HOMO energy levels as compared with the corresponding carbon analogues. The authors are suggested to provide more explanations on these results. Otherwise, it is very difficult for readers to follow.
6. On Page 11, the authors attributed the high (low) PL quantum yield to a large (small) λ_{r} . Particularly, the high PL quantum yield for BO4e was attributed to its ICT characteristics. Such a logic deduction is very puzzling. Indeed, molecular structures determine the properties, and therefore the underlying explanations for PL quantum yields should be provided.
7. The authors have suggested that the higher-lying triplet excited state (T_n) with energy close-lying and matched transition orbital composition with the singlet excited state (S_1) facilitates the ISC of dBO-PAHs, emphasizing the new molecular design perspective. However, the rationality on the presence of the T_n (T_8 for BO1a and T_3 for BO6) and relationship between the $\Delta E_{S_1-T_n}$ and the molecular structure is insufficient, especially when comparing with halogen- and carbonyl-containing UOP materials. For better clarity, besides HOMO and LUMO, the spatial plots of frontier molecular orbitals at the optimized S_0 geometry that mentioned in the main text (HOMO-4, HOMO-5, LUMO+1) are also suggested to

provide.

8. For the performance of OLEDs using dBO-PAHs as the host materials, the brightness, color purity and EQEs have improved when compared with the those reported in the literatures. However, in most cases, the comparisons are not fair, such as different emitters and fabrication conditions.

9. The authors are suggested to pay attention to the writing and typos. On Page 1, the title “Applications in Organic Optoelectronic Materials” should be “Applications as Organic Optoelectronic Materials”. Also, the organic optoelectronic materials in this work only refer to the host materials for OLEDs. On Page 18, “sresults” should be “results”.

Response to the Reviewers' Comments

Reviewer #1 (Remarks to the Author):

In this manuscript, the authors synthesized and characterized a series of dBO-PAHs with low-temperature ultralong afterglow property, and also demonstrated the potential for OLEDs as *n*-type host materials. The results of low-temperature ultralong organic phosphorescence are interesting, but the explanation is unclear and is not suitable in its present form for publication in Nat. Commun.

Correspondence: Many thanks for your recognition and constructive comments.

Major issues

1. In page 5, "For the synthesis of central symmetric double BO-fused pyrenes (Type I), electron-rich alkyl and aryl substitutions on either of the phenyl rings were tolerated in moderate to excellent yields (53-94%), except for BO1g (35%)", and authors mentioned that "it is difficult to attain the second electrophilic borylation on an electron-deficient phenyl ring", after authors introduced somehow electron-donating groups such as methyl and phenyl group (**BO1b**, **BO1c**, and **BO1d**), the reaction yield keeps similar to **BO1a** around 90%, but in the case of **BO1e**, the reaction yield almost goes down to half (just 55%), furthermore, the reaction yield of BO1g is half lower than that of BO1a. So could you please explain the reasons for the lower reaction yield and show the byproducts during the reactions. Also the same question on the reactions of **BO3a**, **BO3b** and **BO3c**.

Correspondence: Thank you for the valuable comment. It is true that, on the basis of electronic effect, the reaction yield of **BO1c**, **BO1e**, **BO1g** and **BO3b** should be similar to those of other BO1 and BO3 series.

Actually, the several series of dBO-PAHs were synthesized in a large timespan, about two years. Considering the high humidity of the air in summer Hangzhou, China, we think that it is the AlCl₃ absorbed moisture during the saving and weighting process? To clarify this question, we purchased new AlCl₃ and carefully re-carried out the reactions, the AlCl₃ was weighted quickly.

About the synthesis of **BO3b**, **BO1g** and **BO1e**:

Because of the available starting materials, **A3b** was synthesized in a large scale. According the optimized reaction condition, when 4 mol% AlCl₃ was used, it is true that a large polar by-product was observed (Fig. R1-1); the title product **BO3b** was isolated in 45% yield, and the by-product was also isolated, however, ¹H NMR showed that there was some impurity in it, resulting in the total purity of the by-product was about 80% on the basis of the ¹H NMR spectrum. Through the analysis of the ¹H NMR spectrum, we speculated it should be an intermediate of 6-mesityl-8-methyl-2-(*p*-tolyl)-6*H*-dibenzo[*c,e*][1,2]oxaborinin-3-ol (**BO3b-OH**) on the basis of the proton signal of 10.17 (s, 1H, OH) and one Mes. To confirm the structure, HRMS was also performed and the [M+H]⁺ 419.2177, found 419.2172 was in good agreement with the chemical structure of **BO3b-OH** (please see SI). Moreover, the single crystal X-ray analysis further confirmed the structure of **BO3b-OH** (Fig. R1-1). Therefore, the isolated yield of **BO3b-OH** was 41%, purity was about 80%.

Then, we tried to transform the intermediate **BO3b-OH** to title product **BO3b**, and it was found that **BO3b** could be isolated in 50% yield in the present of 20mol % AlCl₃ (Fig. R1-2).

Finally, increasing the AlCl₃ to 20mol %, from the starting material **A3b**, the title product **BO3b** could be obtained as a white solid 1.59 g in 92% isolated yield in one-pot protocol (Fig. R1-3).

Similar result was also found in the synthesis of **BO1g**; increasing the AlCl₃ to 20mol %, from the starting material **A1g**, the title product **BO1g** could be obtained as a white solid 1.15 g in 94% isolated yield in one-pot protocol (Fig. R1-4).

For the synthesis of **BO1e** (Fig. R1-5), the reaction also generated an intermediate **Int1e**, which was very difficult to completely isolate from the title product **BO1e**. The chemical structure of **Int1e** was not confirmed by ^1H NMR; we are trying to confirm its structure by X-ray single crystal diffraction, but still not obtain ideal single crystal. However, if the increasing the AlCl_3 to 2.0 equiv, from the starting material **A1e**, the title product **BO1e** could be obtained in 74% isolated yield in one-pot protocol (Fig. R1-5). This might be attributed to the steric hindrance of the meta-position Mes.

On the basis of above results, it was found that for some substrates, more AlCl_3 were needed to promote the complete conversion of the intermediates to title products.

All the above results had been added in the Supporting Information.

Fig. R1-1 (also in SI). Synthesis of **BO3b**

Fig. R1-2 (also in SI). Synthesis of **BO3b** from **BO3b-OH**

Fig. R1-3 (also in SI). Improved synthesis of BO3b

Fig. R1-4 (also in SI). Synthesis of BO1g

Fig. R1-5 (also in SI). Synthesis of BO1e

Fig. R1-6 (also in SI). Improved synthesis of BO2

By the way, for the synthesis of **BO2**, as other reviewer's comment, we also double checked the reason for its relatively low yield (53%), which was because of its poor solubility in the eluent: petroleum ether ~ petroleum ether/dichloromethane = 5:1. If toluene was used to dissolve all the product in the column chromatography, the isolated yield become normal. As the requirement of host for device fabrication, a large scale of was synthesized with 6.71 g in 80% isolated yield (Fig R1-6). Please also see our detailed respond for the question 3 of reviewer 2. Thanks.

We also carried out large scale synthesis of **BO1b**, **BO1c**, **BO2**, and **BO3a**, for their further device fabrication as *n*-type host materials (please also see our respond for question 1 of review 2). They all could be smoothly obtained using 4 mol % AlCl_3 with isolated yields of 80% (9.56 g), 87% (16.30 g), 80% (6.71 g), and 82% (10.98 g), respectively (please see Supporting Information for details).

We have stated in the manuscript of the section "Synthetic Method Development and Structure Characterization" (orange color): "As illustrated in Fig. 3, the one-pot protocol exhibited a broad substrate scope. For the synthesis of central symmetric double BO-fused pyrenes (Type I), electron-rich alkyl and aryl substitutions on either of the phenyl rings were tolerated in moderate to excellent yields (68–94%), except for **BO1g** (35%) however, 2.0 equiv AlCl_3 was needed for **BO1e** to promote the complete conversion of intermediate."

"Importantly, large scale synthesis of **BO1b**, **BO1c**, **BO2** and **BO3a** were also carried out for their further device fabrication as host materials; they all could be smoothly obtained using 4 mol % AlCl_3 with isolated yields of 80% (9.56 g), 87% (16.30 g), 80% (6.71 g) and 82% (10.98 g), respectively (Fig. S3). The developed method can provide versatile DBO-PAHs, enabling the investigation of their potential applications in organic electronics."

In Fig.3, "The obtained product weight was provided in parentheses. ^a20 mol% AlCl_3 was used. ^bA by-product 6-mesityl-8-methyl-2-(*p*-tolyl)-6*H*-dibenzo[*c,e*][1,2]oxaborinin-3-ol (**BO3b-OH**, CCDC: 2290997) was also isolated in 41% yield. ^c2.0 equiv AlCl_3 was used."

Additionally, to avoid the impact of moisture on the reaction, we also have stated in the Supporting Information of

the section “General Experimental Procedure for the Synthesis of Dimethoxy Compounds and Double BO-Fused Polycyclic Aromatic Hydrocarbons (dBO-PAHs) (orange color): “... ..The mixture was stirred at room temperature for 18–40 hours monitoring by TLC until the demethylation was completely. Then AlCl₃ was added quickly (The AlCl₃ is highly hygroscopic and should be weighed quickly!), the flask was placed in oil bath (75 °C) and stirred for 8–10 hours,... ..”.

2. Some dBO-PAHs (like BO3d, BO3e and BO4e) exhibited well resolved vibronic emission spectra, but some dBO-PAHs (like BO1f, BO1g and BO2) exhibited unstructured and broad emission spectra, the authors should explain the differences of molecular vibrations.

Correspondence: Thank you for the comment and suggestion. Typically, several factors affect the emission spectra of the dBO-PAHs. (I) The coupling between the electrons and vibrations of the dBO-PAHs. (II) The intrinsic structures of the dBO-PAHs, including molecular rigidity, polarity, and symmetry. (III) Structural deformation of excited states, and number of metastable conformations in excited states. (IV) Solvent effect, that is the interaction between the excited-state dBO-PAHs molecules and the solvent molecules; typically, the emission spectra would be more unstructured and broader in solvent with higher polarity.

The broadening of the emission spectra is caused by electronic vibronic coupling between the S₀ and excited states as well as by structural relaxation at the excited states. From the DFT calculations (Figs. S15–S19), we can see that the HOMO and LUMO of the dBO-PAH compounds are primarily localized between atoms, thus forming π-bonds. This bonding/antibonding character enhances the interactions between the electronic and nuclear vibrational motion between the S₀ and excited states as well as the vibrational relaxation at the excited states; moreover, there are also some metastable conformations in their excited states. These reasons result in broad emission spectra with several peaks. However, **BO2** possesses asymmetric molecular structure, which is different from most of the Type I dBO-PAHs, Type III, Type IV, Type V and Type VI dBO-PAHs possessing symmetry molecular structures. The asymmetric **BO2** should have more metastable conformations in excited states than those of other symmetric dBO-PAHs, thus **BO2** exhibits unstructured and broad emission spectra. Actually, **BO1f** and **BO1g** have substituents (Ph for **BO1f**, and Mes for **BO1g**) on the molecular core, but the substituent is far from the two steric Mes groups; this enable the substituents Ph and Mes to be easier to vibrate or rotate, and **BO1f** and **BO1g** have many asymmetric molecular conformations, and also have more metastable conformations in excited states. The electronic vibronic coupling of the dBO-PAHs developed in this work is much different with the BN-PAHs reported in the literatures; the BN-PAHs have the multiple resonance effect of the boron and nitrogen atoms induces the localization of the HOMO and LUMO on different atoms, which can minimize their bonding/antibonding character; the resulting non-bonding molecular orbitals (MOs) minimize the electronic vibronic coupling and vibrational relaxation, allowing the realization of an extremely sharp PL band. [Kondo, Y., Yosshiura, K., Kitera, S., Nishi, H., Oda, S., Gotoh, H., Sasada, Y., Yanai, M., Hatakeyama, T. Nature Photonics **13**, 678–682 (2019)].

We have stated in the manuscript of the section “Photophysical Properties” (orange color): “Most dBO-PAHs exhibited well resolved vibronic emission spectra, and dBO-PAHs with large π-conjugation showed one dominant emission peak with small FWHM value, like **BO3d**, **BO3e** and **BO4e**; by contrast, **BO1f**, **BO1g** and **BO2** displayed unstructured and broad emission spectra, resulting in large FWHM values, this should be attributed to their relatively strong electronic vibronic coupling between the S₀ and excited states as well as by structural relaxation at the excited states.”.

3. The authors mentioned about the high PLQY (95%) of BO4e was attributed to the ICT characteristics, but other dBO-PAHs such as BO4a and BO4b also possess strong ICT characteristics from UV-Vis absorption spectra and showed much lower PLQY of 62% for BO4a and 50% for BO4b, respectively. The detailed explanation should be described.

Correspondence: Thank you for the comment and suggestion. The oscillator strength (f) has great effect on the PLQY, large f facilitates the increase of the PLQY; and f is also closely related to the molecular structures. Typically, symmetric and extended linear conjugated molecules with ICT characteristics are beneficial to the improvement of the f .

In order to explain the relationship between f and PLQY (Φ_{PL}), we have added the detailed explanation in the Supporting Information of the section "Estimation of experimental oscillator strength, and the relationship with k_r and Φ_{PL} " as follows (orange color):

Estimation of experimental oscillator strength, and the relationship with k_r and Φ_{PL} . [Turro, N. J., Ramamurthy, V., Scaiano, J. C. Modern Molecular Photochemistry of Organic Molecules; University Science Books; pp195–200, (2009).]

$$\Phi_{PL} = k_r / (k_r + k_{nr}) \quad (S1)$$

$$k_r = \nu^2 f \quad (S2)$$

Where the Φ_{PL} is photoluminescence efficiency, k_r is radiative rate, k_{nr} is non-radiative rate, ν is the wavenumber corresponding to the maximum wavelength of absorption, f is oscillator strength. From the **Equation S1** and **Equation S2**, we can see that the f plays an critical role in the increase of Φ_{PL} .

The theoretical quantity of the oscillator strength f in the classical theory of light absorption is related to the extinction coefficient ϵ of absorption by the expression:

$$f \equiv 4.3 \times 10^{-9} \int \epsilon \, d\bar{\nu} \quad (S3)$$

Where ϵ is the experimental extinction coefficient and $\bar{\nu}$ is the energy of the absorption. With the assumption that the absorption spectrum is a smooth Gaussian curve which can be approximated by an isosceles triangle, we can have $\int \epsilon \, d\bar{\nu} \sim \epsilon_{\max} \Delta\bar{\nu}_{1/2}$, where ϵ_{\max} is the value of ϵ at the absorption maximum and $\Delta\bar{\nu}_{1/2}$ (in cm^{-1}) is the full-width half-maximum (FWHM) of the absorption band.

With the experiment data extracted from the UV-Vis spectrum and **Equation S3**, the approximate of f can be obtained as **Equation S4**:

$$f \sim 4.3 \times 10^{-9} \epsilon_{\max} \Delta\bar{\nu}_{1/2} \quad (S4)$$

Therefore, the f of **BO4a**, **BO4b** and **BO4e** can be calculated as:

$$f(\text{BO4a}) = (4.3 \times 10^{-9}) \times (2.95 \times 10^4) \times [(1/(348.5 \times 10^{-7}) - (1/(357.5 \times 10^{-7}))] = 0.092.$$

$$f(\text{BO4b}) = (4.3 \times 10^{-9}) \times (4.15 \times 10^4) \times [(1/(352.5 \times 10^{-7}) - (1/(361.0 \times 10^{-7}))] = 0.12.$$

$$f(\text{BO4e}) = (4.3 \times 10^{-9}) \times (7.57 \times 10^4) \times [(1/(379.0 \times 10^{-7}) - (1/(390.0 \times 10^{-7}))] = 0.24.$$

Moreover, the increase of the f from **BO4a**, **BO4b** and **BO4e** can be explained by the relationship between the classical concept of oscillator strength and the quantum mechanical transition dipole moment (**Equation S5**).

$$\text{Oscillator strength } f \propto \mu_i^2 = (er)^2 \quad \text{Transition dipole moment} \quad (S5)$$

Where μ_i is the induced transition dipole moment (or dipole strength) corresponding to electronic transition (absorption or emission). The dipole strength of a transition may be set equal to er , which can be viewed as the average size of the transition dipole, where r is the dipole length. By combining the classical oscillator strength with the quantization of oscillation of electrons, we have the expression relating f and μ_i , which is given by **Equation S6**,

$$f = \left(\frac{8\pi m_e \bar{\nu}}{3he^2} \right) \mu_i^2 \cong 10^{-5} \bar{\nu} |er_i|^2 \quad (S6)$$

The f is a function of the μ_i , where m_e is the mass of electron, $\bar{\nu}$ is the energy of the transition (in cm^{-1}), h is

Planck's constant. The μ_j and r should be much larger for the delocalized ICT transition in **BO4e** than those in **BO4a** and **BO4b**, this is because **BO4e** with two fluorenyl units possesses longer linear conjugation system compared to **BO4a** and **BO4b**.

On the basis of the above analyses, we also revised the manuscript in the section of “Photophysical Properties” (orange color):

“Additionally, the experimentally calculated oscillator strength (f) of S_1 - S_0 for **BO4e** was estimated to be 0.24 (see supporting information for details), the high f was attributed to the C_2 symmetry of the molecular geometry, and extended linear conjugation system by two fluorenyl units, as well as the incorporation of the BO units, which could induce a larger transition dipole moment compared to no BO-fused PAHs.^{29,30}”

“In particular, **BO4a** and **BO4b** had relatively low Φ_{PL} of 62% and 50%, respectively; **BO4e** showed an ultra high Φ_{PL} of 95%, together with a short τ_f of 1.4 ns, leading to a large k_r of $6.8 \times 10^8 \text{ s}^{-1}$ and a small k_{nr} of only $0.4 \times 10^8 \text{ s}^{-1}$; in contrast, (Table 1). The high Φ_{PL} of **BO4e** was attributed to the ICT characteristics, reflecting the importance of the BO incorporation; and also **BO4e** had larger f (0.24) than those of **BO4a** ($f = 0.092$) and **BO4b** ($f = 0.12$) (see supporting information for details).”

4. In page 13, “All the above factors endow **BO1a** with a long τ_p (4.0 s)”, but in the Fig.6a, τ_p of **BO1a** is 3.1 s, this makes me confused about the explanation why **BO1a** showed low-temperature UOP (6.0 s) but **BO1d** showed two times longer lifetime (20 s) than that of **BO1a**. The calculations of natural transition orbitals and spin-orbital coupling are necessary to discuss intersystem crossing process.

Correspondence: Thank you for the comment and suggestion. We are so sorry for the mistake, it should be “All the above factors endow **BO1a** with a long t_p (3.1 s)”, which was in agreement with the data in Fig. 6a and Table 1.

As the reviewer's suggestion, natural transition orbitals and spin-orbital coupling were also calculated for **BO1a**, **BO1d** and **BO6** (see Fig. 6h, 6i, and also Fig R1-7 above). Considering comprehensively the factors of energy gaps, transition orbital composition, and spin-orbit coupling between the excited states, **BO1a** should possess multiple efficient pathways from S_1 to T_n , they were S_1 to T_8 , S_1 to T_7 , and S_1 to T_5 . In contrast, **BO1d** and **BO6** had only one efficient pathways from S_1 to T_7 and S_1 to T_2 , respectively. Moreover, although **BO1d** had small $\Delta E_{S_1-T_7}$ of 0.021 eV, the $\xi(S_1-T_7)$ was very tiny (0.24 cm^{-1}), which resulted in slow ISC and relatively long UOP.

On the basis of the above calculations, we also revised the manuscript in the section of “Photophysical Properties” (orange color): “To obtain a deep understanding of the UOP properties of the dBO-PAHs, TD-DFT calculations were performed on their excited state levels and some main transition configurations (Fig. 6h–6i, S47–S51). As illustrated in Fig. 6h, **BO1a** had an extremely small S_1 - T_8 energy gap ($\Delta E_{S_1-T_8}$) of -0.023 eV , along with the same transition orbital composition of 97% HOMO→LUMO for S_1 and 73% HOMO→LUMO for T_8 , facilitating an efficient ISC from S_1 to T_8 ; TD-DFT calculations also showed that spin-orbit coupling (SOC) values of $\xi(S_1-T_7)$ and $\xi(S_1-T_5)$ were very large, 1.01 and 1.61 cm^{-1} , respectively, together with small ΔE_{ST} values of 0.185 and 0.052 eV, revealing that the ISC from S_1 to T_7 and T_5 were also efficient pathways according the Fermi's golden rule. Moreover, natural transition orbital (NTO) analyses indicated that ISC from S_1 to T_8 was complete local excited (LE) character, and ISC from S_1 to T_7 had hybridized local and charge transfer (HLCT) characters in the molecular core (Fig. S46). Then, the internal conversion (IC) from T_8 , T_7 , T_5 , to T_1 can occur rapidly, due to several small energy gaps separating them. In addition, the large $\Delta E_{S_1-T_1}$ of 0.386 eV and the mismatched transition orbital compositions of T_1 and S_1 prevent the reverse ISC from T_1 to S_1 . Moreover, the rigid molecular geometry of **BO1a** (Fig. S5) and the surrounding solid environment suppress the nonradiative decay. All the

above factors endow **BO1a** with a long τ_p (3.1 s). Similarly, **BO6** exhibited an even smaller $\Delta E_{S_1-T_2}$ of only 0.013 eV, the same transition orbital composition for S_1 (99% HOMO→LUMO) and T_2 (97% HOMO→LUMO) and large $\xi(S_1-T_2)$ of 0.93 cm^{-1} (Figure 6i); the ISC from S_1 to T_2 also showed complete LE character in the molecular core (Fig. S46). In contrast, **BO1d** had small $\Delta E_{S_1-T_7}$ of 0.021 eV, but the $\xi(S_1-T_7)$ was very tiny (0.24 cm^{-1}), this resulted in slow ISC and relatively long UOP (Fig. S46). These results provide a new perspective for the molecular design and development of metal- and halogen-free, single-component, dual-emission materials with UOP.”.

Fig. R1-7 (also as Fig. 6h, 6i in manuscript and Fig. S47 in Supporting Information). Theoretical calculation. TD-DFT calculated singlet and triplet energy levels, main transition configurations, spin-orbit coupling (SOC) values and natural

transition orbital (NTO) analyses of **BO1a**, **BO1d** and **BO6** at B3LYP/6-31G(d) level based on optimized S_0 geometry. Selected frontier orbital distributions and energy levels are also illustrated.

5. The rigid molecular structure of dBO-PAHs and the solid state at 77K low temperature suppress the nonradiative decay, and the neat film of dBO-PAHs is also a solid state, so how about the UOP property of dBO-PAHs in the film state at room temperature or the ice point?

Correspondence: Thank you for the comment and valuable question. Actually, we had carefully investigated the UOP property of the dBO-PAHs in the polymethyl methacrylate (PMMA) films, and no UOP was observed. I think this could be attributed that, at room temperature in the PMMA films, the dopant molecules are not completely stationary, they could also have dipole orientation induced by the surrounding molecules; this would facilitate the nonradiative decay. Actually, exciton- and polaron-induced reversible dipole reorientation in amorphous organic semiconductor films had been investigated by Zhang's group [Deng, C., Zhang, L., Wang, D., Tsuboi, T., Zhang, Q. *Adv. Optical Mater.* **7**, 1801644 (2019).]. As the suggestion, we also tested the UOP property at ice point, and no UOP was observed, neither.

We have clarified the results in the manuscript of the section "Ultralong Low-Temperature Phosphorescence" (orange color): "Notably, no UOP property was observed for the dBO-PAHs in PMMA films at room temperature or the ice point.".

6. dBO-PAHs based host material achieved high luminance such as device 7 (15722 cd/m²), the authors should explain the reason why this kind of host material can achieve such high luminance.

Correspondence: Thank you for the comment and suggestion. Compared to the devices using only *p*-type hosts mCP (device 2), mCBP (device 3) or *n*-type host **BO1b** (device 4), device 7 using co-host **BO1b**:mCBP showed significant L_{\max} and EQE_{\max} enhancements, this is attributed to the improved charge balance in the emitting layer (EML), and also the reduction of charge accumulation at the interface between the electron-blocking layer (EBL) and EML or EML and hole-blocking layer (HBL).

We have clarified the reason in the manuscript of the section "Performance of dBO-PAHs as Host Materials for High-Brightness Deep-Blue OLEDs" (orange color): "Surprisingly, **BO1b**:mCBP-based device 7 also achieved an L_{\max} value of up to 15722 cd/m², this was attributed the improved charge balance in the EML and the reduction of charge accumulation at the interface between the EBL/EML or EML/HBL.".

7. In Fig. 7 e, h, and k, the EQE-L plots showed zigzag-shape, this is possible to obtain an underestimated or overestimated results, the authors should recheck the device performances.

Correspondence: Thanks for the comment. It was true that the EQE curves of some devices showed zigzag-sharp at low luminance regions, this was because a wide-range luminance meter was used in the brightness measurements. To avoid underestimated or overestimated the results, we fitted the data and rechecked the device performances in Table 2 in the manuscript (Fig. R1-8). Thanks.

Fig. R1-8 (also as Fig. S57 in Supporting Information). EL properties of deep-blue OLEDs. a, b Fitted EQE vs. luminance plots.

8. For blue phosphorescence emitters, OLED device stability is also very important, how about the device operation lifetime based on dBO-PAHs host materials.

Correspondence: Thank you for your comment. It is true that the device operation lifetime one of the important requirements for their potential application in display. However, many factors could great affect the device operational lifetime, such as the chemical stability of the functional materials, the purities of the materials, device structures, the atmosphere of device fabrication and so on; especially for the deep-blue OLEDs, the operational lifetime is a well-known bottleneck issue in OLED field and has not been widely reported, probably due to the great challenge of realizing good operational lifetime for the deep-blue OLEDs, although great progress had been made in 2022 by Samsung [*Nat. Photonics* **16**, 212 (2022)] on the basis of our previous report [*Adv. Mater.* **26**, 7116–7121 (2014); *Adv. optical Mater.* **3**, 390 (2015)], however, great challenge still remain to simultaneously realize high EQE, high colour-purity, high maximum luminance and long operational lifetime in the same device.

As the reviewer’s suggestion, the operational lifetimes of device 1 (PtON1:26mCPy) and newly fabricated device 10 (PtON1:BO1c:mCBP), device 11 (PtON1:BO1g:mCBP), device 14 (PtON-TBBI:BO2:mCBP), device 15 (PtON-TBBI:BO3a:mCBP) and device 14 (PtON-TBBI:BO6:mCBP) were measured (Fig. R1-9). They exhibited operational lifetimes of LT_{50} with $L_0 = 500 \text{ cd/m}^2$ at the range of 0.2–1.0 h, which was mainly because of the poor stability of DPEPO. However, it was also found that the co-hosts-based OLEDs with improved charge balance increased the operational lifetime. Moreover, although PtON1 ($E_{T1} = 2.82 \text{ eV}$) had higher E_{T1} than that of PtON-TBBI ($E_{T1} = 2.78 \text{ eV}$), PtON1-based devices 10 and 11 also demonstrated significantly longer operational lifetimes compared to PtON-TBBI-based devices 14, 15 and 16, which revealing that the CIE_y value also had great effect on the operational lifetime.

We have added the discussion in the manuscript of the section “Performance of dBO-PAHs as Host Materials for High-Brightness Deep-Blue OLEDs” (orange color): “Last, the operational lifetimes of some deep-blue OLEDs were measured (Fig. S59); because of the poor stability of DPEPO, they showed LT_{50} with $L_0 = 500 \text{ cd/m}^2$ of 0.2–1.0 h. However, PtON1:BO1c:mCBP device 10 and PtON1:BO1g:mCBP device 11 exhibited obviously longer operational lifetimes than

those of PtON1:26mCPy device 1, this revealed that the co-hosts-based OLEDs with improved charge balance increased the operational lifetime. Moreover, although PtON1 ($E_{T1} = 2.82$ eV)⁴⁸ had higher E_{T1} than that of PtON-TBBI ($E_{T1} = 2.78$ eV), PtON1-based devices 10 and 11 also demonstrated significantly longer operational lifetimes compared to PtON-TBBI-based devices 14, 15 and 16, which revealing that the CIE_y value also had great effect on the operational lifetime.”.

Fig. R1-9 (also as Fig. S60 in Supporting Information). Operational lifetimes of the deep-blue OLEDs ($L_0 = 500$ cd/cm²)

9. Ref 32 in Table 2 is a wrong citation. I guess this is the correct citation “Sun, J., Ahn, H., Kang, S. et al. Exceptionally stable blue phosphorescent organic light-emitting diodes. *Nat. Photon.* 16, 212–218 (2022)”

Correspondence: Thank you for the revision. Yes, the old Ref 32 was “Sun, J., Ahn, H., Kang, S. et al. Exceptionally stable blue phosphorescent organic light-emitting diodes. *Nat. Photon.* 16, 212–218 (2022)”. The old Ref 32, 33, and 34 in Table 2 were incorrectly numbered, which should be new Ref 46, 47, and 48, respectively. We have revised them. Thanks.

Thanks again for your valuable comments and suggestions, I hope that our responds could address your comments.

Reviewer #2 (Remarks to the Author):

G. Li and Y. She et al. reported a series of double BO-doped PAHs by a one-pot borylation of OMe-substituted precursors. The skeletons could be edited by changing the connection of aryl groups and the positions of methoxyl groups. They also applied the dBO-PAHs as host materials for OLED device, which showed high efficiency. The authors have demonstrated potential borylation mechanism. The structures were well characterized, and the photophysical properties were well recorded and elucidated by theoretical studies. For these reasons, the reviewer thinks this work is quite interesting. In this context one also has to ask why the authors compared their EQE with Pt(II)-based compounds? The reported materials/molecules are organic materials, however, Pt(II)-based compounds are coordinated compounds. They do not belong to the same class of comparators. How about their EQE compared to the BN series of compounds (BN-PAHs or NBN-PAHs, such as: *Adv. Optical Mater.* 2022, 10, 2201714)? Is there any advantages compared with these BN-PAHs or NBN-PAHs for their BO-PAHs (such as the FWHM, Φ_{PL} , CIE, etc.)? The newly synthesized compounds indisputably possess an obvious structural beauty and as such already the synthetic achievement itself is of considerable significance. Nevertheless, at the same time all these arguments in my opinion do not sufficiently justify publication of the manuscript in its present form in *Nature Communications*. Moreover, the manuscript suffers from several additional issues which need to be addressed prior to resubmission. To sum up, the quality of the manuscript is not yet sufficiently high for publication, and I recommend the manuscript to be rejected for now. After appropriate adjustments and the additional experiments as suggested below, it should be reconsidered as a new manuscript.

Correspondence: Many thanks for your recognition and constructive questions. I hope that our responds below could address your comments. Thanks.

I agree that the dBO-PAHs and Pt(II)-based compounds do not belong to the same class of comparators.

The reasons compared with Pt(II)-based compounds. *The first reason*, actually, for the devices in the section of “Performance of dBO-PAHs as Host Materials for High-Brightness Deep-Blue OLEDs”, the newly developed dBO-PAHs in this work were employing as host materials, which did not emit light in the emitting layer of the devices; the Pt(II) complexes, e.g. PtON1, PtON7-dtb and PtON-TBBI in this manuscript, were used as emitters. Moreover, all the fabricated devices were deep-blue OLEDs with $CIE_y < 0.2$. This was one reason that we compared their EQE with the previously reported deep-blue OLEDs employing similar Pt(II)-based compounds as emitters. Moreover, this work demonstrated that the host materials with rational molecular design could significant enhance the device performances of deep-blue OLEDs. *Another reason*, in the past decade, great progress had been made for deep-blue OLEDs [*Adv. Mater.* **26**, 7116–7121 (2014), efficient and high color-purity deep-blue OLED, but large efficiency roll-off and small maximum luminance; *Nature Photonics*, **16**, 212 (2022), efficient and stable deep-blue OLED, but low color-purity and small maximum luminance], however, great challenge still remain to simultaneously realize high EQE, high color-purity, high maximum luminance and long operational lifetime in the same device. The device performances of Pt(II)- and Ir(III)-base deep-blue OLEDs, including structures of emitter, emitter/host, CIE coordinates, λ_{EL} , L_{max} , EQE at max/100/1000 cd/m^2 , reference/published year, were summarized in Supporting Information (Fig. S61, S62, Table S7, S8). These information could provide good references about the advantages and disadvantages of this work and previously reported work. *The third reason*, because *Nature Communication* was a comprehensive journal, that has many readers in different fields, the comparison enables the readers to have an intuitive and emotional knowledge about the

development about deep-blue OLEDs; and this will be beneficial for attracting more readers.

About device performances compared with BN-PAHs or NBN-PAHs. *First of all*, it is indisputable that BN-PAHs or NBN-PAHs with multiple resonance (MR) effect are exciting emissive molecules with outstanding photophysical properties. *Especially*, they could realize ultra-narrow emission spectra with FWHM values less than 20 nm, which enabled the devices to show high color-purity, and realized deep-blue OLEDs with CIE_y even less 0.1. *Additionally*, by judicious molecular design, they could also possess high Φ_{PL} near to 100%, which also enabled the peak EQE nearly 40%. [Adv. Mater. **28**, 2777 (2016); Nature Photonics, **13**, 678 (2019);... for reviews, see Adv. Optical Mater. **10**, 2201714 (2022); Adv. Sci. 2303504 (2023); Adv. Funct. Mater. 2306880 (2023); J. Mater. Chem. C **11**, 6471 (2023)] **Most importantly**, the BN-PAHs or NBN-PAHs with multiple resonance (MR) effect have been successfully commercial application as blue emitter in display for smartphone because of their high-color purity and high stability. **However**, there is still an issue, the BN-PAHs or NBN-PAHs used in the commercial display demonstrated low EQEs, only about 10%, revealing that the BN-PAHs or NBN-PAHs used as *fluorescent emitters*, and only 25% singlet excitations were harvest to convert into light, and the 75% triplet excitations not yet utilized. **In contrast**, because of the heavy-atom effect induced strong spin-orbital coupling (SOC), Pt(II) complexes could harvest all the electrongenerated singlet and triplet excitations, and realize unity internal quantum efficiency (IQE), although the reported maximum EQE still did not exceed 30%, however, it has been demonstrated that the EQEs of Pt(II)-based deep-blue OLEDs fabricated in industrial production could exceed 25%. Moreover, Pt(II)-based deep-blue OLEDs also realized narrow emission spectra with FWHM values about 20 nm (device 9 and devices 14–16 in this work), high peak EQEs and high L_{max} . Pt(II)-based phosphorescent emitters have been recognized as the critical candidate to realize efficient and stable deep-blue OLEDs for commercial display, and Pt(II) blue emitters has become a hot research topic in academia and industry in the past five years.

Additionally, we also have summarized the deep-blue emitter BN-PAHs and NBN-PAHs, and also their device performances, which were illustrated in Fig. S63, Fig. S64 and Table S9 in the Supporting Information. The deep-blue OLEDs developed in this work still maintain top performances compared to BN-PAHs and NBN-PAHs-based OLEDs, or even the hyperfluorescent OLEDs.

(1) For the device part, why did the authors only apply **BO1b** for the device application, how about other BO-PAHs? For sure, compared to pure-carbon **CC1b**, **BO1b** has the advantages, but the authors did not give any explanation why they did not use other type of BO-PAHs in their manuscript.

Correspondence: Thank you for your comment. Actually, the evaporator resources were not enough to fabricated the devices for all the dBO-PAHs. Moreover, blue, especially, deep-blue OLEDs are important. In order to be employed as host materials for deep-blue OLEDs, their T_1 energy levels (E_{T1}) should be high enough. **BO1b** was selected as candidate for the potential application as host for deep-blue OLEDs, which was because the large E_g , high E_{T1} , together with the high thermal stability; more importantly, enough **BO1b** were available for further purification and sublimation.

As the suggestion, we also selected some other dBO-PAHs with high E_{T1} as potential hosts for deep-blue OLEDs, such as other Type I compounds **BO1c** ($E_{T1} = 3.02$ eV), **BO1g** ($E_{T1} = 3.01$ eV), Type II compound **BO2** ($E_{T1} = 2.88$ eV), Type III compound **BO3a** ($E_{T1} = 2.88$ eV), and Type VI compound **BO6** ($E_{T1} = 2.81$ eV).

Then, the selected dBO-PAHs were synthesized in large scales to make enough materials for further purification and sublimation. They were all successfully synthesized according our optimized reaction condition, and they newly obtained dBO-PAHs were characterized by ^1H NMR, the purities were characterized by HRMS after sublimation (see

Table. R2-1 below). All the ^1H NMR and HPLC spectra were provided in the Supporting Information. Their purity was by HPLC analyses

As expected, **BO1c** ($E_{\text{T1}} = 3.02$ eV), **BO1g** ($E_{\text{T1}} = 3.01$ eV) with high E_{T1} could also be as *n*-Type host materials for PtON1-based deep-blue OLEDs, achieving peak EQEs of 27.8% and 25.1%, retained high EQEs of 20.0% and 19.2% at 1000 cd/m^2 , respectively; and they also achieved high L_{max} s of 27219 and 22683 cd/m^2 , respectively; the L_{max} s were greatly increased compared to **BO1b**-based device 7 (15722 cd/m^2); the L_{max} of device 10 (27219 cd/m^2) represented the record-high L_{max} among Pt(II)-based deep-blue OLEDs with $\text{CIE}_y < 0.20$. By comparison, **BO2** ($E_{\text{T1}} = 2.88$ eV) and **BO3a** ($E_{\text{T1}} = 2.88$ eV) are not suitable for PtON1-based deep-blue OLEDs; PtON1:**BO2** device 12 and PtON1:**BO3a** device 13 exhibited significantly low EQEs and L_{max} s; this was attributed to the similar E_{T1} of **BO2** (2.88 eV) and **BO3a** (2.88 eV) compared to PtON1 (2.82 eV). However, **BO2** and **BO3a**, as well as **BO6** ($E_{\text{T1}} = 2.81$ eV) could be acted as hosts for PtON-TBBI ($E_{\text{T1}} = 2.78$ eV, Fig. S58)-based deep-blue OLEDs, devices 14, 15 and 16 showed narrow EL spectra with FWHM values of 21–22 nm, they also realized peak EQEs of 22.2%, 19.6% and 21.7%, and high L_{max} s of 14481, 15765 and 10071 cd/m^2 , respectively. (see Table. R2-2 and Fig. R2-1 below)

We also have added some discussion in the manuscript of the section “Performance of dBO-PAHs as Host Materials for High-Brightness Deep-Blue OLEDs” (orange color): “Then, other dBO-PAHs with high E_{T} as hosts in deep-blue OLEDs were also investigated (Fig. 8, S58, Table 2). **BO1c** ($E_{\text{T1}} = 3.02$ eV)-based device 10 and **BO1g** ($E_{\text{T1}} = 3.01$ eV)-based device 11 using PtON1 as emitter also demonstrated good device performances with peak EQEs of 27.8% and 25.1%, retained high EQEs of 20.0% and 19.2% at 1000 cd/m^2 , respectively; and they also achieved high L_{max} s of 27219 and 22683 cd/m^2 , respectively; the L_{max} s were greatly increased compared to device 7 (15722 cd/m^2); the L_{max} of device 10 (27219 cd/m^2) represented the record-high L_{max} among Pt(II)-based deep-blue OLEDs with $\text{CIE}_y < 0.20$ (Table S5),^{46–49,55–63} By comparison, PtON1:**BO2** device 12 and PtON1:**BO3a** device 13 exhibited significantly low EQEs and L_{max} s; this was attributed to the similar E_{T1} of **BO2** (2.88 eV) and **BO3a** (2.88 eV) compared to PtON1 (2.82 eV⁴⁸), which could not enable efficient energy transfer from host to emitter molecules. However, **BO2** and **BO3a**, as well as **BO6** ($E_{\text{T1}} = 2.81$ eV) could be acted as hosts for PtON-TBBI ($E_{\text{T1}} = 2.78$ eV, Fig. S59)-based deep-blue OLEDs, devices 14, 15 and 16 showed narrow EL spectra with FWHM values of 21–22 nm, they also realized peak EQEs of 22.2%, 19.6% and 21.7%, and high L_{max} s of 14481, 15765 and 10071 cd/m^2 , respectively. Last, the operational lifetimes of some deep-blue OLEDs were measured (Fig. S60); because of the poor stability of DPEPO, they showed LT_{50} at $L_0 = 500$ cd/m^2 of 0.2–1.0 h. However, PtON1:**BO1c**:mCBP device 10 and PtON1:**BO1g**:mCBP device 11 exhibited obviously longer operational lifetimes than those of PtON1:26mCPy device 1 and PtON-TBBI-based devices 14, 15 and 16, this revealed that the co-hosts-based OLEDs with improved charge balance increased the operational lifetime, and the CIE_y value also had great effect on the operational lifetime. The device performance should be further improved by optimizing the ratio of *n*-type dBO-PAHs and *p*-type host in the EML and employing advanced functional materials of other layers.”.

(2) 2D NMR spectra of **BO5** might be provided.

Correspondence: Thank you for the suggestion. 2D NMR spectra of ^1H - ^1H correlated spectroscopy (COSY) (see Fig. R2-2), heteronuclear single quantum coherence (HSQC) (see Fig. R2-3) and heteronuclear multiple-bond correlation (HMBC) spectra (see Fig. R2-4) were performed, and all the protons and carbons were assigned on the basis of the above spectra. Moreover, single crystal of **BO5** was carefully grown and analyzed by X-ray diffraction (see Fig. R2-5), and C–H $\cdots\pi$ interactions between Mes and BO core in **BO5** were also observed.

We also have stated in the manuscript of the section “Crystallographic Analyses”, and also added some revision

(orange color): “To further confirm the molecular structures of the dBO-PAHs and determine their molecular geometries, single crystals of **BO1a**, **BO2**, **BO3a**, **BO3d**, **BO4b**, **BO4d**, **BO4e**, **BO5**, and **BO6** were carefully grown and analyzed by X-ray diffraction.”, “Moreover, the B–C bond lengths of the BO-cores in **BO1a**, **BO2**, **BO4d**, **BO4e**, **BO4e**, **BO5**, and **BO6** ranged between 1.522 and 1.549 Å; Moreover, the B–C bond lengths of the BO-cores in **BO1a**, **BO2**, **BO4b**, **BO4d**, **BO4e**, **BO5**, and **BO6** ranged between 1.522 and 1.549 Å;”, “and C–H··· π interactions between fluorenyl moieties were also observed in **BO4e** (Fig. S12) and between Mes and BO core in BO5 (Fig. S13) were also observed.”.

(3) Type I, Type V and Type VI introduce two boron atoms to the same benzene rings. It seemed reasonable that Type II product was formed owing to the electronic effects. But the first borylation exclusively took place at the less steric methoxyl group?

Correspondence: Thank you for the valuable comment, this is a good question. The isolated yield for the synthesis of **BO2** was not very high (53%) in our first manuscript; and in order to double check is there other by-products to generate and also preparation more material for device fabrication, we re-performed this reaction in a large scale (see Fig. R2-6 below). From the TLC monitoring, it was found there was only one intermediate after the adding of BBr₃, and also nearly only one borylation product was observed, and the isolated yield was increased to 80% (6.71 g). The above results reveal that the first borylation was dominated by the steric effect, and the second borylation was attributed to the electronic effect. Notably, **BO2** has a poor solubility in the eluent: petroleum ether ~ petroleum ether/dichloromethane = 5:1, and toluene was used to dissolve all the product in the column chromatography, we have intentionally added explanation and reminder in the Supporting Information. The low isolated yield (53%) in our first manuscript was because only partial product was dissolved and separated.

We have stated in the manuscript of the section “Synthetic Method Development and Structure Characterization”, and also added some revision (orange color): “Interestingly, upon adjusting the position of one phenyl group (Ar²) from 1,3-dimethyl-2,5-diphenylbenzene (**A1f**) to 1,3-dimethyl-2,4-diphenylbenzene (**A2**), the asymmetric dBO-benzo[*c*]chrysene **BO2** (Type II), bearing two boron atoms on different phenyl rings, could be exclusively separated, and the corresponding Type I compound was not observed; this revealed that the first borylation was dominated by the steric effect, and the second electrophilic borylation had a high regioselectivity and preferentially occurred on an electron-rich phenyl (Fig. 1b).”.

We also have stated in the Supporting Information of the section “Synthetic Method Development and Structure Characterization”, and also added some revision (orange color): “... ..to afford **BO2** (eluent: eluent: petroleum ether ~ petroleum ether/dichloromethane = 5:1~ toluene) as a white solid 6.71 g in 80% yield. (Note: **BO2** has a very poor solubility, and toluene was used to dissolve all the product in the column chromatography.)”.

(4) The precursors with two methoxyl groups connected to the central benzene rings (*meta*-, or *para*-) could afford dBO-PAHs smoothly. Is it possible to further applied the one-pot strategy to the *ortho*-analog?

Correspondence: Thank you for the valuable comment, this is a good question. Actually, the *ortho*-analog was not performed because the starting material 1,4-dibromo-2,3-dimethoxybenzene or 1,4-dichloro-2,3-dimethoxybenzene were not commercially available. Therefore, two similar substrates 1,4-dibromo-2,3-dimethoxy-5,6-dimethylbenzene (**A7a**) and 2,3-dimethoxy-1,4-diphenylnaphthalene (**A7b**) were synthesized, and they were also characterized by ^1H NMR and ^{13}C NMR (please see Supporting Information). Then they were carried out with our standard reaction condition with 4 mol% AlCl_3 as catalyst. However, substrate **A7a** gave messy result, a series of products were observed, and no desired products were isolated; then, catalyst AlCl_3 was increased to 2.0 equiv, and just afforded similar results. (see Fig. R2-7 below). Substrate **A7b** also gave similar result in the present of 2.0 equiv AlCl_3 (see Fig. R2-8 below). We think that this might be attributed to the *ortho*-dimethoxy, which was vulnerable to generate borate ester and its derivatives. On the basis of above results, the developed one-pot method was not suitable for the synthesis of the *ortho*-analog.

All the above results were added to Supporting Information.

We also have stated in the manuscript of the section "Synthetic Method Development and Structure Characterization", and also added some revision (orange color): "*Ortho*-analog was also tried to synthesized by using the precursors of 1,4-dibromo-2,3-dimethoxy-5,6-dimethylbenzene (**A7a**) and 2,3-dimethoxy-1,4-diphenylnaphthalene (**A7b**), but they gave messy results (see Supporting information for details).".

Thanks again for your valuable comments and suggestions, I hope that our responds could address your comments.

Table R2-1 (also in SI). The purification of dBO-PAHs for their further used as *n*-Type host materials.

sample	sublimation temperature (°C)	vacumm (10^{-5} Torr)	crude sample (g)	sublimated sampled (g)	yield (%)	purity (%)
BO1b	40/100/130/150/250/250/270	0.16	2.5	2.1	84	99.90
BO1c	40/100/130/150/250/250/270	5.0	6.4	6.0	94	99.56
BO2	40/100/130/150/250/250/280	5.0	3.4	3.0	88	99.68
BO3a	40/100/130/150/250/250/280	5.6	6.1	5.5	90	99.94
BO6	130/150/160/200/240/240/260	5.0	0.6	0.53	88	99.28

HPLC Analysis Conditions: Column: Masiall® C18-BIO, 5 μm , 250 \times 4.6 mm; Mobile phase: methanol/tetrahydrofuran = 95/5(v/v); flow rate: 1.0 mL/min; Abs. detector: 254 nm.

Table R2-2 (also as Table 2 in manuscript)Tabl. Summary of device performance for Pt(II)-based deep-blue OLEDs and the comparison with representatively reported literatures.

Device	EML	Voltage ^[a] [V]	λ_{PL} [nm]	FWHM [nm]	CIE(x, y)	EQE ^[b] [%]	L_{max} [cd/m ²]
device 1	8% PTON1:92% 26mCPy	3.8/5.6	454	57	(0.143, 0.159)	18.3/17.9/13.7/9.6	15271
device 2	8% PTON1:92% mCP	4.0/5.8	462	56	(0.138, 0.171)	18.5/17.9/16.3/12.0	10284
device 3	8% PTON1:92% mCBP	3.8/6.2	461	55	(0.139, 0.167)	16.3/16.0/15.4/11.8	11740
device 4	8% PTON1:92% BO1b	4.0/6.9	453	50	(0.138, 0.134)	22.8/21.5/16.3/8.8	8007
device 5	8% PTON1:46% BO1b :46% 26mCPy	3.9/6.2	454	50	(0.138, 0.130)	24.8/24.4/20.9/15.0	12982
device 6	8% PTON1:46% BO1b :46% mCP	3.8/6.0	460	52	(0.138, 0.154)	22.1/20.0/19.2/14.1	13098
device 7	8% PTON1:46% BO1b :46% mCBP	3.5/6.2	456	51	(0.138, 0.142)	27.1/25.2/21.8/15.6	15722
device 8	8% PTON7-dtb:46% BO1b :46% mCBP	3.4/7.4	453	28	(0.138, 0.088)	27.6/20.0/16.0/7.9	5670
device 9	8% PTON-TBBI:46% BO1b :46% mCBP	3.4/6.5	458	21	(0.134, 0.104)	28.0/22.8/19.0/13.4	9377
device 10	8% PTON1:46% BO1c :46% mCBP	3.1/5.5	456	56	(0.141, 0.168)	27.8/24.3/20.0/14.4	27219
device 11	8% PTON1:46% BO1g :46% mCBP	3.8/5.9	458	56	(0.140, 0.170)	25.1/21.9/19.2/14.7	22683
device 12	8% PTON1:46% BO2 :46% mCBP	4.4/7.6	462	64	(0.146, 0.208)	17.3/17.2/14.5/9.2	9867
device 13	8% PTON1:46% BO3a :46% mCBP	4.9/7.9	462	61	(0.143, 0.194)	15.6/15.3/14.1/9.6	9803
Device 14	8% PTON-TBBI:46% BO2 :46% mCBP	3.2/5.8	458	21	(0.135, 0.103)	22.2/18.0/14.9/10.4	14481
Device 15	8% PTON-TBBI:46% BO3a :46% mCBP	3.8/6.1	459	22	(0.134, 0.107)	19.6/17.0/14.8/10.8	15765
Device 16	8% PTON-TBBI:46% BO6 :46% mCBP	3.9/6.7	459	22	(0.133, 0.111)	21.7/16.2/13.4/8.7	10071
Ref. 46	13% PtON-BBI:27% SiTrzCz2:60% SiCzCz	2.8/6.3	455	48	(0.141, 0.197)	25.4/24.6/23.4/---	2000
Ref. 47	6% PTON1: 94% 26mCPy	3.9/6.3	454	47	(0.15, 0.13)	25.2/23.3/16.8/---	1000
Ref. 48	6% PTON7-dtb:47% PO15:47% TAPC	2.8/4.4	451	29	(0.148, 0.079)	24.8/22.7/11.0/---	1555

[a] Voltage at 1 and 1000 cd/m². [b] Maximum EQE, and EQEs at 100, 1000 and 5000 cd/m².

Fig R2-1 (also as Fig. 8). Device performances.

BO5-18mg CDC13 0620

Fig R2-2 (also in Supporting Information). ¹H-¹H COSY spectrum of BO5.

BO5-18mg CDC13 0620

Fig R2-3 (also in Supporting Information). HSQC spectrum of BO5.

Fig R2-4 (also in Supporting Information). HMBC spectrum of BO5.

Fig R2-5 (also as Fig. S13 in Supporting Information). Single crystal structure. X-ray single crystal diffraction structure and crystal packing of **BO5**. Hydrogen atoms were omitted for clarity. Ellipsoids are shown at the 50% probability level.

Fig R2-6 (also in SI). Improved synthesis of BO2

Fig R2-7 (also in SI). Attempt for the synthesis of BO7a

Fig R2-8 (also in SI). Attempt for the synthesis of BO7b

Reviewer #3 (Remarks to the Author):

In this manuscript, Li, She and coworkers developed a one-pot tandem demethylation-electrophilic substitution reactions for double-oxygen-fused polycyclic aromatic hydrocarbons (dBO-PAHs). Using this method, six types of dBO-PAHs have been successfully prepared and fully characterized by NMR, HRMS and X-ray analysis. Particularly, the incorporation of the B-O units endows dBO-PAHs with low-temperature ultralong afterglow of up to 20 s after UV excitation, which is a very interesting phenomenon that has received a lot of attention nowadays. Moreover, the potential applications of dBO-PAHs as host materials for metal complex-based deep-blue OLEDs have also been demonstrated. Despite a lot of works have been done, there are still some parts remained to be well clarified and the significance/inspiration of this work is not very great. Considering the high requirements of this journal, major revisions are needed after consideration.

Correspondence: Many thanks for your constructive comments and valuable suggestions. I hope that our responds below could address your comments. Thanks.

More detailed comments are shown below:

1. The authors have claimed that the incorporation of B-O units enables the dBO-PAHs with unique optoelectronic properties, such as high fluorescence quantum yield in dichloromethane solution and single-component, low-temperature ultralong afterglow. However, the description on the structure-property logic is confusing. For example, the authors claimed that the low-aromaticity of BO-fused rings lead to high thermal and chemical stabilities (Page 2) as well as high quantum yields (Page 4) of dBO-PAHs. In fact, the good stabilities of dBO-PAHs are due to the synergistic effects of the p-pi conjugation, the delocalization of the lone electron pairs of the oxygen atoms as well as the bulky Mes moieties. Therefore, the authors are suggested to pay more attention to the logic flow.

Correspondence: Thank you for the valuable comment and suggestion. We are sorry for the confused statement, and I agree with the reviewer's suggestion.

We have revised it in the manuscript of the section "Introduction" (orange color): "Theoretical and experimental studies revealed that the incorporation of the B-O units into the PAHs greatly affected their aromaticity and electronic properties. The synergistic effects of the p- π conjugation, the delocalization of the lone electron pairs of the oxygen atoms as well as the bulky Mes moieties (Mes: 2,4,6-trimethylphenyl) enabled the dBO-PAHs to possess good stabilities. The dBO-PAHs showed strong fluorescence emission with quantum efficiencies (Φ_{PL}) of up to 95% at room temperature (RT) in dichloromethane; moreover, most dBO-PAHs exhibit single-component, dual-emission with both nanosecond fluorescence and second-level phosphorescence at 77 K in 2-methyltetrahydrofuran (2-MeTHF), and the persistent phosphorescence enabled the dBO-PAHs to exhibit an ultralong low-temperature afterglow of up to 20 s after UV excitation."

2. Continuing the comment 2, comparing the optoelectronic properties of structurally related all carbon-based PAHs (such as CC1b) and BN-fused dBO-PAHs with those of dBO-PAHs in this work should be made.

Correspondence: Thank you for the comment. Actually, all the carbon-based PAHs in this work were served as model compounds, which were compared with the newly developed dBO-PAHs by theoretical calculations. So, I am sorry that the optoelectronic properties of the carbon-based PAHs are incomplete. However, to have a deep understanding about the differences of the carbon-based PAHs, and BN-fused dBO-PAHs with those of dBO-PAHs in this work, the comparison

of Pyrene, BO1b and previously reported BN1 and DABNA-1 is illustrated in Fig. R3-1.

Fig. R3-1 (also as Fig. S1 in Supporting Information). Theoretical calculation, chemical structures and photophysical properties of Pyrene, BN1, BO1b and DABNA-1. (a) Comparison of calculated frontier orbital distributions, energy levels/gaps, calculated and experimental E_{T1} , MO character, aromaticity. (b, c) Absorption and fluorescence spectrum of Pyrene at RT and 77K.¹ (d) Fluorescence spectrum of BN1 at RT.² (e) Photophysical properties of BO1b developed in this work. (f, g) Emission spectrum of DABNA-1 at RT and 77K.³ [Ref. 1. Turro, N. J., Ramamurthy, V., Scaiano, J. C.

Modern Molecular Photochemistry of Organic Molecules; University Science Books; pp217–226, (2009). Ref. 2. Bosdet, M. J. D. Piers, W. E., Sorensen, T. S., Parvez, M. *Angew. Chem. Int. Ed.* **46**, 4940–4943 (2007). Ref. 3. Hatakeyama, T., Shiren, K., Nakajima, K., Nomura, S., Nakatsuka, S., Kinoshita, K., Ni, J., Ono, Y., Ikuta, T. *Adv. Mater.* **28**, 2777–2781 (2016).]

Molecular orbital and photophysical properties. Typically, the HOMO and LUMO of Pyrene and BN1 are primarily localized between atoms, thus forming π -bonds; this bonding/antibonding character enhances the interactions between the electronic and nuclear vibrational motion between the S_0 and S_1 states as well as the vibrational relaxation at the S_1 state, thus Pyrene and BN1 showed broad and structural emission spectra (Fig. R3-1b, 1c, 1d). In contrast, **BO1b** had less bonding character and exhibited relatively narrow and less structural fluorescence spectrum at RT (Fig. R3-1e). For DABNA-1, the multiple resonance (MR) effect of the boron and nitrogen atoms induces the localization of the HOMO and LUMO on different atoms and minimizes their bonding/antibonding character; the resulting non-bonding molecular orbitals (MOs) minimize the vibronic coupling and vibrational relaxation in the molecule, allowing the realization of an extremely sharp PL spectrum at RT (Fig. R3-1f).

About the E_{T1} , MO character, aromaticity. Pyrene, and the heterocycles of BN1 and DABNA-1 have strong aromaticity, resulting in small energy gaps between HOMO and LUMO, as well as small E_{T1} , therefore, they can not be employed as host materials for deep-blue OLEDs. However, because of the weakened π -conjugation of the BO core, **BO1b** had a significantly higher calculated E_{T1} (3.10 eV) than **CC1b** (2.11 eV), which was also in good agreement with the experimental value of 3.00 eV). The large E_g , high E_{T1} , together with the high thermal stability of **BO1b** enable it to serve as a host material for deep-blue OLED. Moreover, planar BN1 also showed excimer emission (510–550 nm) in 5.19 μ M cyclohexane as well as in single crystal because of strong π -stacking, this was very unfavourable to serve as host material, therefore, bulk group, like Mes were need for the design of dBO-PAHs.

We have added some discussion in the section “Introduction” (orange color): “Polycyclic aromatic hydrocarbons (PAHs), especially heteroatom-embedded PAHs, have attracted considerable attention¹ because of their unique catalytic activity and photophysical/photoelectronic properties, enabling extensive and promising applications in catalysis² and organic electronics, such as organic photovoltaics (OPVs)³, organic field-effect transistors (OFETs)^{4,5}, organic phototransistors (OPTs)⁶, and organic light-emitting diodes (OLEDs)^{7–11}. However, because of strong electronic vibronic coupling and aromaticity, PAHs typically show broad structural emission spectra and small T_1 energy levels (E_{T1}) (Fig. S1).⁹ Among the various main-group elements used as dopants, boron is particularly attractive, owing to its sp^2 hybridization and the presence of the vacant p_z orbital; these features enable the efficient regulation of the molecular skeletons and π -conjugation of the boron-embedded PAHs (B-PAHs), as well as the further tuning of their excited-state properties and charge-transfer characteristics, in order to develop a variety of novel functional materials.^{1,7–13} Especially, boron/nitrogen-embedded PAHs (BN-PAHs) with rational molecular design demonstrated extremely narrowband Gaussian-type emission spectra because of the minimized vibronic coupling by multiple resonance (MR) effect (Fig. S1).^{7–9} However, compared with the widely studied B-PAHs and BN-PAHs,^{1,6–17} scarce data are available on PAHs containing B–O unit (BO-PAHs), and in particular on their optoelectronic properties, due to their inefficient synthetic methods.”

3. On Page 5, the yield of BO1g is relatively low as compared with those of other central symmetric double BO-fused pyrenes (Type I). It seems like the relatively low yield is not due to the electronic effect or the steric effect, so the authors are suggested to give reasonable explanations on this.

Correspondence: Thank you for the valuable comment. I agree with your comment that, on the basis of electronic effect, the reaction yield of **BO1g** should be similar to those of **BO1a**, **BO1b** and **BO1c**. Similarly, the isolated yield of **BO3b** was also abnormal. Reviewer 1 also gave similar comment. In order to clarify the reason for the low isolated yields of some dBO-PAHs, a lot of experiments and analysis characterization works were also conducted, please see below (similar to our respond to the comment 1 of reviewer 1).

Actually, the several series of dBO-PAHs were synthesized in a large timespan, about two years. Considering the high humidity of the air in summer Hangzhou, China, we think that it is the AlCl_3 absorbed moisture during the saving and weighting process? To clarify this question, we purchased new AlCl_3 and carefully re-carried out the reactions, the AlCl_3 was weighted quickly.

About the synthesis of **BO3b**, **BO1g** and **BO1e**:

Because of the available starting materials, **A3b** was synthesized in a large scale. According the optimized reaction condition, when 4 mol% AlCl_3 was used, it is true that a large polar by-product was observed (Fig. R3-2); the title product **BO3b** was isolated in 45% yield, and the by-product was also isolated, however, ^1H NMR showed that there was some impurity in it, resulting in the total purity of the by-product was about 80% on the basis of the ^1H NMR spectrum. Through the analysis of the ^1H NMR spectrum, we speculated it should be an intermediate of 6-mesityl-8-methyl-2-(*p*-tolyl)-6*H*-dibenzo[*c,e*][1,2]oxaborinin-3-ol (**BO3b-OH**) on the basis of the proton signal of 10.17 (s, 1H, OH) and one Mes. To confirm the structure, HRMS was also performed and the $[\text{M}+\text{H}]^+$ 419.2177, found 419.2172 was in good agreement with the chemical structure of **BO3b-OH** (please see SI). Moreover, the single crystal X-ray analysis further confirmed the structure of **BO3b-OH** (Fig. R3-2). Therefore, the isolated yield of **BO3b-OH** was 41%, purity was about 80%.

Then, we tried to transform the intermediate **BO3b-OH** to title product **BO3b**, and it was found that **BO3b** could be isolated in 50% yield in the present of 20mol % AlCl_3 (Fig. R3-3).

Finally, increasing the AlCl_3 to 20mol %, from the starting material **A3b**, the title product **BO3b** could be obtained as a white solid 1.59 g in 92% isolated yield in one-pot protocol (Fig. R3-4).

Similar result was also found in the synthesis of **BO1g**; increasing the AlCl_3 to 20mol %, from the starting material **A1g**, the title product **BO1g** could be obtained as a white solid 1.15 g in 94% isolated yield in one-pot protocol (Fig. R3-5).

For the synthesis of **BO1e** (Fig. R3-6), the reaction also generated an intermediate **Int1e**, which was very difficult to completely isolate from the title product **BO1e**. The chemical structure of was **Int1e** not confirmed ^1H NMR; we are trying to confirm its structure by X-ray single crystal diffraction, but still not obtain ideal single crystal. However, if the increasing the AlCl_3 to 2.0 equiv, from the starting material **A1e**, the title product **BO1e** could be obtained in 74% isolated yield in one-pot protocol (Fig. R3-6). This might be attributed to the steric hindrance of the meta-position Mes.

On the basis of above results, it was found that for some substrates, more AlCl_3 were needed to promote the complete conversion of the intermediates to title products.

All the above results had been added in the Supporting Information.

Fig. R3-2 (also in Supporting Information). Synthesis of BO3b

Fig. R3-3 (also in Supporting Information). Synthesis of BO3b from BO3b-OH

Fig. R3-4 (also in Supporting Information). Improved synthesis of BO3b

Fig. R3-5 (also in Supporting Information). Synthesis of BO1g

Fig. R3-6 (also in Supporting Information). Synthesis of BO1e

Fig. R3-7 (also in Supporting Information). Improved synthesis of BO2

By the way, for the synthesis of **BO2**, as other reviewer's comment, we also double checked the reason for its relatively low yield (53%), which was because of its poor solubility in the eluent: petroleum ether ~ petroleum ether/dichloromethane = 5:1. If toluene was used to dissolve all the product in the column chromatography, the isolated yield become normal. As the requirement of host for device fabrication, a large scale of was synthesized with 6.71 g in 80% isolated yield (Fig R3-7). Please also see our detailed respond for the question 3 of reviewer 2. Thanks.

We also carried out large scale synthesis of **BO1b**, **BO1c**, **BO2**, and **BO3a**, for their further device fabrication as *n*-type host materials (please also see our respond for question 1 of review 2). They all could be smoothly obtained using 4 mol % AlCl_3 with isolated yields of 80% (9.56 g), 87% (16.30 g), 80% (6.71 g), and 82% (10.98 g), respectively (please see SI for details).

We have stated in the manuscript of the section "Synthetic Method Development and Structure Characterization" (orange color): "As illustrated in Fig. 3, the one-pot protocol exhibited a broad substrate scope. For the synthesis of central symmetric double BO-fused pyrenes (Type I), electron-rich alkyl and aryl substitutions on either of the phenyl rings were tolerated in moderate to excellent yields (68–94%), except for **BO1g** (35%) however, 2.0 equiv AlCl_3 was needed for **BO1e** to promote the complete conversion of intermediate."

"Importantly, large scale synthesis of **BO1b**, **BO1c**, **BO2** and **BO3a** were also carried out for their further device fabrication as host materials; they all could be smoothly obtained using 4 mol % AlCl_3 with isolated yields of 80% (9.56 g), 87% (16.30 g), 80% (6.71 g) and 82% (10.98 g), respectively (Fig. S3). The developed method can provide versatile DBO-PAHs, enabling the investigation of their potential applications in organic electronics."

In Fig.3, "The obtained product weight was provided in parentheses. ^a20 mol% AlCl_3 was used. ^bA by-product 6-mesityl-8-methyl-2-(*p*-tolyl)-6*H*-dibenzo[*c,e*][1,2]oxaborinin-3-ol (**BO3b-OH**, CCDC: 2290997) was also isolated in 41% yield. ^c2.0 equiv AlCl_3 was used."

Additionally, to avoid the impact of moisture on the reaction, we also have stated in the Supporting Information of

the section “General Experimental Procedure for the Synthesis of Dimethoxy Compounds and Double BO-Fused Polycyclic Aromatic Hydrocarbons (dBO-PAHs) (orange color): “... ..The mixture was stirred at room temperature for 18–40 hours monitoring by TLC until the demethylation was completely. Then AlCl₃ was added quickly (The AlCl₃ is highly hygroscopic and should be weighed quickly!), the flask was placed in oil bath (75 °C) and stirred for 8–10 hours,... ..”.

4. On page 7, the almost planar BO-fused cores of dBO-PAHs are supported by the sum of degrees of one C-B-C and two C-B-O bonds, not three C–B–C bonds?

Correspondence: Thank you for your comment. We are so sorry for the mistake. Yes, it was not three C–B–C bonds.

We have revised it in the manuscript of the section “Crystallographic Analyses” (orange color): “All the dBO-PAHs had almost planar BO-fused cores, supported by the fact that the sums of the one C–B–C and two C–B–O bond angles around the boron atoms were almost 360.0° for all the dBO-PAHs, revealing the sp² hybridization of the central boron atoms.”.

5. As revealed by DFT calculation results, dBO-PAHs had similar LUMO energy levels and remarkably low-lying HOMO energy levels as compared with the corresponding carbon analogues. The authors are suggested to provide more explanations on these results. Otherwise, it is very difficult for readers to follow.

Correspondence: Thank you for your comment and suggestion. As the suggestion, the frontier orbital distributions, energy levels of all the carbon-based PAHs were added to the Fig. S16, Fig. S17, Fig. S18, and Fig. S19 (please also see Fig R3-8, Fig R3-9, Fig R3-10 and Fig R3-11 below).

From Fig R3-8, we can see that, the HOMO distributions of all CC1 series are nearly identical and they localized on the molecular cores; however, for BO1 series, the distributions are delocalized to peripheral Mes or Ph groups, resulting in reduced electron cloud densities and remarkably low-lying HOMO energy levels. And similar phenomenon was also observed for **BO6** vs **CC6** (Fig R3-9).

From Fig R3-10, we can see that, a large portion of HOMO distributions of BO3 series are on the boron atoms and the BO-fused cores, therefore the electron deficient property of the boron stabilizes the HOMO levels, and making they have low-lying HOMO energy levels. Similar phenomenon was also observed for **BO4a**, **BO4b**, and also in **BO2** and **BO5**; vs **CC6** (Fig R3-11, Fig R3-9); in contrast, because of the extended π -conjugation, the HOMO distributions on the boron atoms in **BO4c**, **BO4d**, **BO4e** significantly reduced, thus, the HOMO level differences between **BO4c**, **BO4d**, **BO4e** and their corresponding **CC4c**, **CC4d**, **CC4e** are also significantly reduced (Fig R3-11).

On the basis of the above analyses, we have added the explanations in the revised manuscript of the section “Theoretical Analyses” (orange color): “... ..however, they possessed remarkably stabilized highest occupied molecular orbitals (HOMOs), resulting in significantly large HOMO–LUMO energy gaps (E_g), this was attributed to the delocalization effect for BO1 series and BO6, and the electron deficient property of the boron for BO2, BO3 series, BO4 series and BO5 (Fig. S15–S18); the LUMO, HOMO, and E_g values were –1.47, –5.28, and 3.81 eV for CC1a vs. –1.50, –6.06, and 4.56 eV for **BO1a**, respectively (Table S5)”.

Fig. R3-8 (also as Fig. S16 in Supporting Information). Theoretical calculation. Comparison of frontier orbital distributions, energy levels, and NICS(1) values of dBO-PAHs **BO1a**, **BO1b**, **BO1c**, **BO1d**, **BO1e** and **BO1g**, as well as their corresponding carbon-based PAHs **CC1a**, **CC1b**, **CC1c**, **CC1d**, **CC1e** and **CC1g**. The calculations were performed at B3LYP/6-31G(d) level.

Fig. R3-9 (also as Fig. S17 in Supporting Information). Theoretical calculation. Comparison of frontier orbital distributions, energy levels, and NICS(1) values of dBO-PAHs BO1a, BO2, BO3a, BO4a, BO5 and BO6, as well as their corresponding carbon-based PAHs CC1a, CC2, CC3a, CC4a, CC5 and CC6. The calculations were performed at B3LYP/6-31G(d) level.

Fig. R3-10 (also as Fig. S18 in Supporting Information) Theoretical calculation. Comparison of frontier orbital distributions, energy levels, and NICS(1) values of dBO-PAHs **BO3a**, **BO3b**, **BO3c**, **BO3d** and **BO3e**, as well as their corresponding carbon-based PAHs **CC3a**, **CC3b**, **CC3c**, **CC3d** and **CC3e**. The calculations were performed at B3LYP/6-31G(d) level.

Fig. R3-11 (also as Fig. S19 in Supporting Information). Theoretical calculation. Comparison of frontier orbital distributions, energy levels, and NICS(1) values of dBO-PAHs **BO4a**, **BO4b**, **BO4c**, **BO4d** and **BO4e**, as well as their corresponding carbon-based PAHs **CC5a**, **CC5b**, **CC5c**, **CC5d** and **CC4e**. The calculations were performed at B3LYP/6-31G(d) level.

6. On Page 11, the authors attributed the high (low) PL quantum yield to a large (small) k_r . Particularly, the high PL quantum yield for BO4e was attributed to its ICT characteristics. Such a logic deduction is very puzzling. Indeed, molecular structures determine the properties, and therefore the underlying explanations for PL quantum yields should be provided.

Correspondence: Thank you for the comment and valuable suggestion.

Typically, the oscillator strength (f) has great effect on the PLQY, large f facilitates the increase of the PLQY; and f is also closely related to the molecular structures. Typically, symmetric and extended linear conjugated molecules with ICT characteristics are beneficial to the improvement of the f .

In order to explain the relationship between f and PLQY (Φ_{PL}), we have added the detailed explanation in the Supporting Information of the section "Estimation of experimental oscillator strength, and the relationship with k_r and Φ_{PL} " as follows (orange color):

Estimation of experimental oscillator strength, and the relationship with k_r and Φ_{PL} . [Turro, N. J., Ramamurthy, V., Scaiano, J. C. Modern Molecular Photochemistry of Organic Molecules; University Science Books; pp195–200, (2009).]

$$\Phi_{PL} = k_r / (k_r + k_{nr}) \quad (S1)$$

$$k_r = \nu^2 f \quad (S2)$$

Where the Φ_{PL} is photoluminescence efficiency, k_r is radiative rate, k_{nr} is non-radiative rate, ν is the wavenumber corresponding to the maximum wavelength of absorption, f is oscillator strength. From the **Equation S1** and **Equation S2**, we can see that the f plays an critical role in the increase of Φ_{PL} .

The theoretical quantity of the oscillator strength f in the classical theory of light absorption is related to the extinction coefficient ϵ of absorption by the expression:

$$f \equiv 4.3 \times 10^{-9} \int \epsilon d\bar{\nu} \quad (S3)$$

Where ϵ is the experimental extinction coefficient and $\bar{\nu}$ is the energy of the absorption. With the assumption that the absorption spectrum is a smooth Gaussian curve which can be approximated by an isosceles triangle, we can have $\int \epsilon d\bar{\nu} \sim \epsilon_{max} \Delta\bar{\nu}_{1/2}$, where ϵ_{max} is the value of ϵ at the absorption maximum and $\Delta\bar{\nu}_{1/2}$ (in cm^{-1}) is the full-width half-maximum (FWHM) of the absorption band.

With the experiment data extracted from the UV-Vis spectrum and **Equation S3**, the approximate of f can be obtained as **Equation S4**:

$$f \sim 4.3 \times 10^{-9} \epsilon_{max} \Delta\bar{\nu}_{1/2} \quad (S4)$$

Therefore, the f of **BO4a**, **BO4b** and **BO4e** can be calculated as:

$$f(\text{BO4a}) = (4.3 \times 10^{-9}) \times (2.95 \times 10^4) \times [(1/(348.5 \times 10^{-7}) - (1/(357.5 \times 10^{-7})))] = 0.092.$$

$$f(\text{BO4b}) = (4.3 \times 10^{-9}) \times (4.15 \times 10^4) \times [(1/(352.5 \times 10^{-7}) - (1/(361.0 \times 10^{-7})))] = 0.12.$$

$$f(\text{BO4e}) = (4.3 \times 10^{-9}) \times (7.57 \times 10^4) \times [(1/(379.0 \times 10^{-7}) - (1/(390.0 \times 10^{-7})))] = 0.24.$$

Moreover, the increase of the f from **BO4a**, **BO4b** and **BO4e** can be explained by the relationship between the classical concept of oscillator strength and the quantum mechanical transition dipole moment (**Equation S5**).

$$\text{Oscillator strength } f \propto \mu_i^2 = (e\mathbf{r})^2 \quad \text{Transition dipole moment} \quad (S5)$$

Where μ_i is the induced transition dipole moment (or dipole strength) corresponding to electronic transition (absorption or emission). The dipole strength of a transition may be set equal to $e\mathbf{r}$, which can be viewed as the average size of the transition dipole, where \mathbf{r} is the dipole length. By combining the classical oscillator strength with the quantization of oscillation of electrons, we have the expression relating f and μ_i , which is given by **Equation S6**,

$$f = \left(\frac{8\pi m_e \bar{\nu}}{3he^2} \right) \mu_i^2 \cong 10^{-5} \bar{\nu} |\mathbf{er}_i|^2 \quad (\text{S6})$$

The f is a function of the μ_i , where m_e is the mass of electron, $\bar{\nu}$ is the energy of the transition (in cm^{-1}), h is Planck's constant. The μ_i and \mathbf{r} should be much larger for the delocalized ICT transition in **BO4e** than those in **BO4a** and **BO4b**, this is because **BO4e** with two fluorenyl units possesses longer linear conjugation system compared to **BO4a** and **BO4b**.

On the basis of the above analyses, we also revised the manuscript in the section of “Photophysical Properties” (orange color):

“Additionally, the experimentally calculated oscillator strength (f) of S_1 - S_0 for **BO4e** was estimated to be 0.24 (see supporting information for details), the high f was attributed to the C_2 symmetry of the molecular geometry, and extended linear conjugation system by two fluorenyl units, as well as the incorporation of the BO units, which could induce a larger transition dipole moment compared to non-BO-fused PAHs.^{29,30}”

“In particular, **BO4a** and **BO4b** had relatively low Φ_{PL} of 62% and 50%, respectively; **BO4e** showed an ultra high Φ_{PL} of 95%, together with a short τ_f of 1.4 ns, leading to a large k_r of $6.8 \times 10^8 \text{ s}^{-1}$ and a small k_{nr} of only $0.4 \times 10^8 \text{ s}^{-1}$; in contrast, (Table 1). The high Φ_{PL} of **BO4e** was attributed to the ICT characteristics, reflecting the importance of the BO incorporation; and also **BO4e** had larger f (0.24) than those of **BO4a** ($f = 0.092$) and **BO4b** ($f = 0.12$) (see supporting information for details).”

7. The authors have suggested that the higher-lying triplet excited state (T_n) with energy close-lying and matched transition orbital composition with the singlet excited state (S_1) facilitates the ISC of dBO-PAHs, emphasizing the new molecular design perspective. However, the rationality on the presence of the T_n (T_8 for **BO1a** and T_3 for **BO6**) and relationship between the $\Delta E_{S_1-T_n}$ and the molecular structure is insufficient, especially when comparing with halogen- and carbonyl-containing UOP materials. For better clarity, besides HOMO and LUMO, the spatial plots of frontier molecular orbitals at the optimized S_0 geometry that mentioned in the main text (HOMO–4, HOMO–5, LUMO+1) are also suggested to provide.

Correspondence: Thank you for the comment and suggestion.

As the reviewer's suggestion, the HOMO–4, HOMO–5, LUMO+1 of **BO1a** and HOMO–4, HOMO–5 of **BO6** had been added. Moreover, natural transition orbitals and spin-orbital coupling were also calculated for **BO1a**, **BO1d** and **BO6** were also calculated in order to have a deep understanding of their excited states (see Fig. 6h, 6i, and also Fig R3-12 below). Considering comprehensively the factors of energy gaps, transition orbital composition, and spin-orbit coupling between the excited states, **BO1a** should possess multiple efficient pathways from S_1 to T_n , they were S_1 to T_8 , S_1 to T_7 , and S_1 to T_5 . In contrast, **BO1d** and **BO6** had only one efficient pathways from S_1 to T_7 and S_1 to T_2 , respectively.

As the reviewer's suggestion, natural transition orbitals and spin-orbital coupling were also calculated for **BO1a**, **BO1d** and **BO6** (see Fig. 6h, 6i, and also Fig R3-12 below). Considering comprehensively the factors of energy gaps, transition orbital composition, and spin-orbit coupling between the excited states, **BO1a** should possess multiple efficient pathways from S_1 to T_n , they were S_1 to T_8 , S_1 to T_7 , and S_1 to T_5 . In contrast, **BO1d** and **BO6** had only one efficient pathways from S_1 to T_7 and S_1 to T_2 , respectively. Moreover, although **BO1d** had small $\Delta E_{S_1-T_7}$ of 0.021 eV, the $\xi(S_1-T_7)$ was very tiny (0.24 cm^{-1}), which resulted in slow ISC and relatively long UOP.

Fig. R3-12 (also as Fig. 6h, 6i in manuscript and Fig. S47 in Supporting Information). Theoretical calculation. TD-DFT calculated singlet and triplet energy levels, main transition configurations and natural transition orbital (NTO) analyses of **BO1a**, **BO1d** and **BO6** at B3LYP/6-31G(d) level based on optimized S_0 geometry. Selected frontier orbital distributions and energy levels are also illustrated.

On the basis of the above calculations, we also revised the manuscript in the section of “Photophysical Properties” (orange color): “To obtain a deep understanding of the UOP properties of the dBO-PAHs, TD-DFT calculations were performed on their excited state levels and some main transition configurations (Fig. 6h–6i, S47–S51). As illustrated in Fig. 6h, **BO1a** had an extremely small S_1 - T_8 energy gap ($\Delta E_{S_1-T_8}$) of -0.023 eV, along with the same transition orbital

composition of 97% HOMO→LUMO for S_1 and 73% HOMO→LUMO for T_8 , facilitating an efficient ISC from S_1 to T_8 ; TD-DFT calculations also showed that spin-orbit coupling (SOC) values of $\xi(S_1-T_7)$ and $\xi(S_1-T_5)$ were very large, 1.01 and 1.61 cm^{-1} , respectively, together with small ΔE_{ST} values of 0.185 and 0.052 eV, revealing that the ISC from S_1 to T_7 and T_5 were also efficient pathways according the Fermi's golden rule. Moreover, natural transition orbital (NTO) analyses indicated that ISC from S_1 to T_8 was complete local excited (LE) character, and ISC from S_1 to T_7 had hybridized local and charge transfer (HLCT) characters in the molecular core (Fig. S46). Then, the internal conversion (IC) from T_8 , T_7 , T_5 , to T_1 can occur rapidly, due to several small energy gaps separating them. In addition, the large $\Delta E_{S_1-T_1}$ of 0.386 eV and the mismatched transition orbital compositions of T_1 and S_1 prevent the reverse ISC from T_1 to S_1 . Moreover, the rigid molecular geometry of **BO1a** (Fig. S4) and the surrounding solid environment suppress the nonradiative decay. All the above factors endow **BO1a** with a long τ_p (3.1 s). Similarly, **BO6** exhibited an even smaller $\Delta E_{S_1-T_2}$ of only 0.013 eV, the same transition orbital composition for S_1 (99% HOMO→LUMO) and T_2 (97% HOMO→LUMO) and large $\xi(S_1-T_2)$ of 0.93 cm^{-1} (Figure 6i); the ISC from S_1 to T_2 also showed complete LE character in the molecular core (Fig. S46). In contrast, **BO1d** had small $\Delta E_{S_1-T_7}$ of 0.021 eV, but the $\xi(S_1-T_7)$ was very tiny (0.24 cm^{-1}), this resulted in slow ISC and relatively long UOP (Fig. S46). These results provide a new perspective for the molecular design and development of metal- and halogen-free, single-component, dual-emission materials with UOP.”

8. For the performance of OLEDs using dBO-PAHs as the host materials, the brightness, color purity and EQEs have improved when compared with the those reported in the literatures. However, in most cases, the comparisons are not fair, such as different emitters and fabrication conditions.

Correspondence: Thank you for the comment. I agree with your comment that because of different emitters, different device structures and fabrication conditions, the comparisons of the device performances, to some extent, is unfair. Thus, the statements about the direct comparisons with those reported in the literatures were deleted.

We have revised the manuscript in the section of “Performance of dBO-PAHs as Host Materials for High-Brightness Deep-Blue OLEDs” (orange color): “Brightness is also an important parameter of device performance, and high brightness is an urgent demand for the outdoor applications of display in smartphone. Our previously reported blue OLED employing PtON7-dtb as emitter demonstrated a high color purity and a high peak EQE, but suffered from the issue of large efficiency roll-off and small L_{max} ($\sim 1555 \text{ cd/m}^2$).⁴⁸ Narrow-band blue emitter PtON7-dtb⁴⁸-based device 8 with co-host of **BO1b**:mCBP was also fabricated (Fig. 7i–7k, Table 2). Device 8 exhibited a narrow EL spectrum peaking at 453 nm with a FWHM of 28 nm and a CIE_y of 0.088, and also realized a peak EQE of 27.6% with small efficiency roll-off, and retained high EQEs of 20.0%, 16.0% and 7.9% at 100, 1000 and 5000 cd/m^2 , respectively; device 8 also achieved an L_{max} of 5670 cd/m^2 . This represented similar color purity, but significantly reduced efficiency roll-off, and enhanced a 3.65-fold L_{max} enhancement compared with our previously reported PtON7-dtb-based blue OLED with TAPC:PO15 as co-hosts⁴⁸ (Table 2). Device 8 showed larger efficiency roll-off than that of device 7 with the same device structure due to the slight higher E_{T_1} and longer τ of PtON7-dtb compared to PtON1⁶⁴. PtON-TBBI-based deep-blue OLED demonstrated exceptionally long operational lifetime, but suffered from low color purity ($\text{CIE}_y = 0.197$) and small L_{max} ($\sim 2000 \text{ cd/m}^2$).⁴⁶ Employing PtON-TBBI with short τ of 2.01 μs as emitter⁴⁶, device 9 demonstrated a peak EQE of 28.0%, which also represented the record-high EQE among Pt(II)-based deep-blue OLEDs with $\text{CIE}_y < 0.20$ ^{46–49,55–63} (Tables 2, S5, Fig. 8), and was also among the highest EQE in the reported Ir(III)-based deep blue OLEDs (Table S6).^{65–70} Device 9 also showed narrow EL spectrum with a small FWHM of 21 nm, resulting in a huge color purity ($\text{CIE}_y = 0.104$); moreover, a L_{max} of 9377 cd/m^2 was also achieved (Table 2). importantly, device 9 also demonstrated a huge color purity ($\text{CIE}_y = 0.104$) and 4.6-fold L_{max} (9377 cd/m^2) enhancement compared to the reported PtON-TBBI-based blue OLED in ref. 32 ($\text{CIE}_y =$

~~0.197, $L_{\max} = 2000 \text{ cd/m}^2$) (Table 2).~~

Additionally, the old Fig. 8 (now Fig. 7b) was also revised and removed the solid arrows which was from the literature reported devices, ~~“The solid arrows show the device performance improvement with the same Pt(II) emitter.”~~.

9. The authors are suggested to pay attention to the writing and typos. On Page 1, the title “Applications in Organic Optoelectronic Materials” should be “Applications as Organic Optoelectronic Materials”. Also, the organic optoelectronic materials in this work only refer to the host materials for OLEDs. On Page 18, “sresults” should be “results”.

Correspondence: Thank you so much for your carefully revision. We have revised them as the suggestion.

Thanks again for your valuable comments and suggestions, I hope that our responds could address your comments.

REVIEWERS' COMMENTS

Reviewer #1 (Remarks to the Author):

The manuscript has been appropriately revised in response to the reviewers' comments and can be recommended for publication.

Reviewer #2 (Remarks to the Author):

The authors made a good job improving some smaller points. At the same time however my overall opinion does not change because it relies on the previously described fact, which by definition can not be altered: novelty of synthetic approach is moderate compared to the reported method; there is nothing particularly attractive in photophysics of these compounds.

Moreover, the response is confused to me: "However, there is still an issue, the BN-PAHs or NBN-PAHs used in the commercial display demonstrated low EQEs, only about 10%, revealing that the BN-PAHs or NBN-PAHs used as fluorescent emitters, and only 25% singlet excitations were harvested to convert into light, and the 75% triplet excitations not yet utilized." This statement is confusing. Are these BN-PAHs or NBN-PAHs used as TADF emitters? If so, the 75% triplet excitations could be utilized, in which the MR-BN molecules have reached the EQE up to 40%.

Unfortunately, the authors did not reply to question 4 from my previous comments.

Reviewer #3 (Remarks to the Author):

The authors have made extensive modifications to the manuscript according to the reviewers' comments, and the quality and the inspiration of the work have been greatly improved. Therefore, this work is suggested to be published in Nat. Commun.

Response to the Reviewer 2 and editor's Comments

Reviewer #2 (Remarks to the Author):

The authors made a good job improving some smaller points. At the same time however my overall opinion does not change because it relies on the previously described fact, which by definition can not be altered: novelty of synthetic approach is moderate compared to the reported method; there is nothing particularly attractive in photophysics of these compounds.

Correspondence: We are grateful for your time and comments.

Actually, in order to address the reviewers' comments, we have carried out some new experiments, and added the significant new experimental results to the manuscript and supporting information. The new experiments and results are as follows.

Large scale or improved synthesis, sublimation and purity analysis of dBO-PAHs:

- (1) Large scale synthesis of **BO1b** (9.56 g) in 80% isolated yield; further purified by recrystallization and sublimation in 84% yield; purity, 99.90% by HPLC analysis.
- (2) Large scale synthesis of **BO1c** (16.30 g) in 87% isolated yield; further purified by recrystallization and sublimation in 94% yield; purity, 99.56% by HPLC analysis.
- (3) The isolated yield of **BO1e** was improved from 53% to 74% by using more AlCl₃ (2.0 equiv).
- (4) The isolated yield of **BO1g** was improved from 35% to 94% by using more AlCl₃ (20 mol%).
- (5) Large scale synthesis of **BO2** (6.71 g) in 80% isolated yield; further purified by recrystallization and sublimation in 88% yield; purity, 99.68% by HPLC analysis.
- (6) Large scale synthesis of **BO3a** (10.98 g) in 82% isolated yield; further purified by recrystallization and sublimation in 90% yield; purity, 99.94% by HPLC analysis.
- (7) Experiment showed that intermediate **BO3b-OH** could transform into title compound **BO3b** in 50% isolated yield in the present of 20 mol% AlCl₃.
- (8) The isolated yield of **BO3b** was improved from 27% to 92% by using more AlCl₃ (20 mol%).
- (9) **BO6** was further purified by recrystallization and sublimation in 88% yield; purity, 99.28% by HPLC analysis.
- (10) Attempt for the synthesis of **BO7a** and **BO7b** were conducted, but failed (see Supplementary Information, Pages S36–S39). *These experiments are our respond for your question 4. Thanks.*

Molecular structure characterization:

- (1) ¹H NMR and ¹³C NMR spectra of **A1e** were added to Supplementary Information.
- (2) ¹³C NMR spectrum of **A1g** was added to Supplementary Information.
- (3) The reaction intermediate **BO3b-OH** in the synthesis of **BO3b** was separated and its structure was confirmed by ¹H NMR, HRMS and X-ray single crystal diffraction analysis (CCDC: 2290227, Supplementary Fig. 4, Supplementary Table 2).
- (4) 2D NMR spectra of ¹H–¹H correlated spectroscopy (COSY), heteronuclear single quantum coherence (HSQC) and heteronuclear multiple-bond correlation (HMBC) spectra of **BO5** were performed and added to Supporting Information.
- (5) Single crystal of **BO5** was carefully grown and analyzed by X-ray diffraction (CCDC: 2280610, Supplementary Fig. 4, 13, Supplementary Table 4).
- (6) ¹H NMR and ¹³C NMR spectra of **A7a** were added to Supplementary Information.
- (7) ¹H NMR and ¹³C NMR spectra of **A7b** were added to Supplementary Information.

Theoretical calculation and electrochemical measurement:

- (1) Frontier orbital distributions and energy levels of all carbon-based PAHs were added to Supplementary Information (Supplementary Fig. 16–19).
- (2) Spin–orbit coupling (SOC) values and natural transition orbital (NTO) analyses of **BO1a**, **BO1d** and **BO6** were added to Supplementary Information (Supplementary Fig. 47).
- (3) Frontier orbital distributions and energy levels of dBO-PAHs were added to Fig. S48–S51 of Supplementary Information.
- (4) Electrochemical properties and energy levels of selected dBO-PAHs, **BO1b**, **BO1c**, **BO1g**, **BO2**, **BO3a** and **BO6**, were added to Supplementary Table 6 of Supplementary Information.
- (5) Cyclic voltammogram (CV) of **BO1c**, **BO1g**, **BO2**, **BO3a**, **BO6**, and differential pulse voltammetry (DPV) of **BO2** were added to Supplementary Information (Supplementary Fig. 61). On the basis of those data, their LUMO levels could be obtained.

Photophysical measurement:

- (1) Low-temperature spectrum of PtON-TBBI was measured and added to Supplementary Information (Supplementary Fig. 58), thus, the E_{T1} of could be calculated as 2.78 eV.
- (2) The oscillator strength (f) of **BO4a**, **BO4b** and **BO4e** were calculated and have added to manuscript and Supplementary information.

Device fabrication and characterization:

- (1) Hole-only devices of **BO1c**, **BO1g**, **BO2**, **BO3a** and **BO6**, as well as electron-only devices of **BO1b**, **BO2**, **BO3a** and **BO6** were fabricated and added to Supplementary Information (Supplementary Fig. 54). The experiments showed that they had high hole-transporting abilities, but hardly transported electrons.
- (2) Devices using **BO1c**, **BO1g**, **BO2**, **BO3a** and **BO6** as host materials were also investigated. Device 10 (PtON1:**BO1c**:mCBP), device 11 (PtON1:**BO1g**:mCBP), device 12 (PtON1:**BO2**: mCBP), device 13 (PtON1:**BO3a**:mCBP), device 14 (PtON-TBBI:**BO2**:mCBP), device 15 (PtON-TBBI:**BO3a**:mCBP) and device 14 (PtON-TBBI:**BO6**:mCBP) were fabricated, and their device performances were illustrated in Table 2 and Fig. 8. Better device performances were obtained, for example, device 10 achieved a L_{max} of 27219 cd/m² with a peak EQE of 27.8%, which was further significantly increased compared to the L_{max} of device 7 (15722 cd/m²). These results reveal the universality of the dBO-PAHs as n -type host materials for deep-blue OLEDs.
- (3) Operational lifetimes of deep-blue OLED, device 1, device 10, device 11, device 14, device 15 and device 16 were measured and added to Supplementary Information (Supplementary Fig. 64). The results revealed that this revealed that co-hosts-based OLEDs with improved charge balance increased the operational lifetime, and the CIE_y value also had great effect on the operational lifetime.

Our work mainly involves three aspects: **First**, one-pot strategy for the facile synthesis of boron-oxygen-fused polycyclic aromatic hydrocarbons (dBO-PAHs) was developed, realizing the large-scale and chemically diverse synthesis of dBO-PAHs. **Second**, the dBO-PAHs exhibit low temperature dual emission containing both fluorescence and phosphorescence, as well as ultralong afterglow of up to 20 s. **Third**, rational molecular design enables the dBO-PAHs to possess high E_{T1} over 3.0 eV; thus, they can be employed as n -type host materials, which are critical for the device performance enhancement for deep-blue OLEDs, and they also can be used as n -type hosts for other deep-blue OLEDs using pyrazole and NHC-based tetradentate Pt(II) complexes as emitters, and also show significant device performance

enhancement, we are now preparing a manuscript and will be published in future. **Moreover**, the BO-PAHs can be also as UV-emitters for highly efficient UV-OLEDs, and the work is under review now.

Moreover, the response is confused to me: “However, there is still an issue, the BN-PAHs or NBN-PAHs used in the commercial display demonstrated low EQEs, only about 10%, revealing that the BN-PAHs or NBN-PAHs used as fluorescent emitters, and only 25% singlet excitons were harvest to convert into light, and the 75% triplet excitons not yet utilized.” This statement is confusing. Are these BN-PAHs or NBN-PAHs used as TADF emitters? If so, the 75% triplet excitons could be utilized, in which the MR-BN molecules has reached the EQE up to 40%.

Correspondence: Thanks for the comment. I totally agree that, in previously reported literatures, the BN-PAHs or NBN-PAHs could be used as TADF emitters, and nearly 40% EQE could be achieved.

I am sorry that I might not express myself clearly, I meant that the BN-PAHs had been used as blue emitters in commercial display for smart phone now, but the BN-PAHs-based devices in industrial production showed low CEs (about 10 cd/A) and low EQEs (about 10%), these results revealed that the BN-PAHs still used as fluorescent emitters, might involve a little TTA mechanism.

Unfortunately, the authors did not reply to question 4 from my previous comments.

Correspondence: Thanks for the comment. Maybe something was lost.

Actually, we did perform many experiments to reply the question 4, and the results had been added in our last Respond Letter, as well as in the Resubmitted Manuscript and the Supplementary Information (see Supplementary Information, Pages S36–S39). Attempt for the synthesis of BO7a and BO7b were conducted, but failed. Thanks.

Please see our respond for the question 4 in last respond letter:

(4) The precursors with two methoxyl groups connected to the central benzene rings (*meta*-, or *para*-) could afford dBO-PAHs smoothly. Is it possible to further applied the one-pot strategy to the *ortho*-analog?

Correspondence: Thank you for the valuable comment, this is a good question. Actually, the *ortho*-analog was not performed because the starting material 1,4-dibromo-2,3-dimethoxybenzene or 1,4-dichloro-2,3-dimethoxybenzene were not commercially available. Therefore, two similar substrates 1,4-dibromo-2,3-dimethoxy-5,6-dimethylbenzene (**A7a**) and 2,3-dimethoxy-1,4-diphenylnaphthalene (**A7b**) were synthesized, and they were also characterized by ¹H NMR and ¹³C NMR (please see Supporting Information). Then they were carried out with our standard reaction condition with 4 mol% AlCl₃ as catalyst. However, substrate **A7a** gave messy result, a series of products were observed, and no desired products were isolated; then, catalyst AlCl₃ was increased to 2.0 equiv, and just afforded similar results. (see Fig. R2-7 below). Substrate **A7b** also gave similar result in the present of 2.0 equiv AlCl₃ (see Fig. R2-8 below). We think that this might be attributed to the *ortho*-dimethoxy, which was vulnerable to generate borate ester and its derivatives. On the basis of above results, the developed one-pot method was not suitable for the synthesis of the *ortho*-analog.

All the above results were added to Supporting Information.

We also have stated in the manuscript of the section “Synthetic Method Development and Structure Characterization”, and also added some revision (orange color): “*Ortho*-analogs were also tried to synthesized by using the precursors of 1,4-dibromo-2,3-dimethoxy-5,6-dimethylbenzene (**A7a**) and 2,3-dimethoxy-1,4-diphenylnaphthalene (**A7b**), but they gave messy results (see Supporting information for details).”.

Fig R2-7 (also in SI). Attempt for the synthesis of **B07a**

Fig R2-8 (also in SI). Attempt for the synthesis of **B07b**

Thanks again for your time and comments, I hope that our responds could address your comments.

SUBMISSION INFORMATION

In order to accept your paper, we require the following:

- A revised author checklist describing your response to our editorial requests (attached).

Correspondence: Thank you. We have completed it.

- The final version of your manuscript as a Word or LaTeX file, with all changes highlighted in the text and any tables prepared using the table menu in Word or the table environment in LaTeX.

Correspondence: Thank you. We have completed it as request.

- If using LaTeX, please use numerical references only for citations, and include the references within the manuscript file itself. If you wish to use BibTeX, please copy the reference list from the .bbl file, paste it into the main manuscript .tex file, and delete the associated \bibliography and \bibliographystyle commands.

Correspondence: We did not used LaTeX.

- The complete author list provided in the manuscript file, which must match that given on our manuscript tracking system. The author list in the main manuscript file will be used during typesetting of your article.

Correspondence: Thank you. We have double checked it.

- Production-quality versions of each figure as a separate file containing all panels. To ensure the swift processing of your paper, please provide the highest quality versions of your images and when combining different figure parts into one file for layout, use a vector-based application such as Adobe Illustrator or Microsoft Powerpoint. We recommend .ai, .eps, .pdf, .ppt. Figures divided into panels should be labelled with a lower-case, boldface 'a', 'b', etc. in the top left-hand corner. If resolution is not of sufficient quality, production of your paper will be held whilst replacement files are obtained. For detailed guidance on figure preparation,

see <https://www.nature.com/documents/aj-artworkguidelines.pdf>

Correspondence: Thank you. We revised the paper as request above.

- Please note that we do not modify the text in figures to conform to style during the production process. Please ensure that your figures are presented accurately and adhere to the guidance provided.

Correspondence: OK. Thank you.

- Any updated checklists that verify compliance with our research ethics and data reporting standards in PDF format.

Correspondence: OK. Thank you. We have updated the checklists.

- The final version of the Supplementary Information in one PDF file.

Correspondence: OK. Thank you.

- Any Supplementary Movie, Audio, Data and Software submitted as separate files. Supplementary Data and Source Data must be provided as .xls, .xlsx or .zip files, while Supplementary Software must be supplied as .zip files.

** Please note that we do not edit Supplementary Information files; they must be finalised prior to acceptance of the

paper. **

Correspondence: OK. Thank you.

- If you wish, an interesting image (but not an illustration or schematic) for consideration as a Featured Image on the Nature Communications homepage. The file should be 1200x675 pixels in RGB format and should be uploaded as a Related Manuscript File with the title “featured image suggestion”. In addition to our home page, we may also use this image (with credit) in other journal-specific promotional material. If your featured image is chosen you will need to complete a Licence to Publish form which will be sent to you via DocuSign at a later stage.

Correspondence: Thank you. We did not made it.

OPEN ACCESS

Nature Communications is a fully open access journal. Articles are made freely accessible on publication under a CC BY license (Creative Commons Attribution 4.0 International License). This license allows maximum dissemination and re-use of open access materials and is preferred by many research funding bodies.

Correspondence: OK. Thank you.

For further information about article processing charges, open access funding, and advice and support from Nature Portfolio, please visit <http://www.nature.com/ncomms/about/open-access>

Correspondence: OK. Thank you.

At acceptance, you will be provided with instructions for completing this CC BY license on behalf of all authors. This grants us the necessary permissions to publish your paper.

Correspondence: OK. Thank you.

ORCID

Nature Communications is committed to improving transparency in authorship. As part of our efforts in this direction, we are now requesting that all authors identified as ‘corresponding author’ create and link their Open Researcher and Contributor Identifier (ORCID) with their account on the Manuscript Tracking System (MTS) prior to acceptance. ORCID helps the scientific community achieve unambiguous attribution of all scholarly contributions. For more information please visit <http://www.springernature.com/orcid>

For all corresponding authors listed on the manuscript, please follow the instructions in the link below to link your ORCID to your account on our MTS before submitting the final version of the manuscript. If you do not yet have an ORCID you will be able to create one in minutes.

Correspondence: OK. Thank you. We have completed it.

IMPORTANT: All authors identified as ‘corresponding author’ on the manuscript must follow these instructions. Non-corresponding authors do not have to link their ORCIDs but are encouraged to do so. Please note that it will not be possible to add/modify ORCIDs at proof. Thus, if they wish to have their ORCID added to the paper they must also follow the above procedure prior to acceptance.

To support ORCID's aims, we only allow a single ORCID identifier to be attached to one account. If you have any issues

attaching an ORCID identifier to your MTS account, please contact the Platform Support Helpdesk.

Correspondence: OK. Thank you. We have completed it.

POLICIES

In recognition of the time and expertise our reviewers provide to Nature Communications's editorial process, as of November 2018, we formally acknowledge their contribution to the external peer review of articles published in the journal. All peer-reviewed content will carry an anonymous statement of peer reviewer acknowledgement, and for those reviewers who give their consent, we will publish their names alongside the published article. For more information, please refer to our FAQ page at <https://www.nature.com/documents/ncomms-reviewer-information.pdf>

Correspondence: OK. Thank you.

Nature Portfolio journals encourage authors to share their step-by-step experimental protocols on a protocol sharing platform of their choice. Where such protocols are available, please provide a DOI or other citation details in the paper. Nature Portfolio's *Protocol Exchange* is a free-to-use and open resource for protocols; protocols deposited in *Protocol Exchange* are citable and can be linked from the published article. More details can be found at <https://www.nature.com/protocolexchange/about>.

Correspondence: OK. Thank you.